# Post-lockdown abatement of COVID-19 by fast periodic switching

**Michelangelo Bin**[1], **Peter Y. K. Cheung**[2], **Emanuele Crisostomi**[3], **Pietro Ferraro**[2], **Hugo Lhachemi**[4], **Roderick Murray-Smith**[5], **Connor Myant**[2], **Thomas Parisini**[1,6,7]*, **Robert Shorten**[2,8]*, **Sebastian Stein**[5], **Lewi Stone**[9,10]*

**1** Department of Electrical and Electronic Engineering, Imperial College London, London, United Kingdom, **2** Dyson School of Design Engineering, Imperial College London, London, United Kingdom, **3** Department of Energy, Systems, Territory and Constructions Engineering, University of Pisa, Pisa, Italy, **4** L2S, CentraleSupélec, Gif-sur-Yvette, France, **5** School of Computing Science, University of Glasgow, Glasgow, Scotland, **6** Department of Engineering and Architecture, University of Trieste, Trieste, Italy, **7** KIOS Research and Innovation Center of Excellence, University of Cyprus, Nicosia, Cyprus, **8** Department of Electrical and Electronic Engineering, University College Dublin, Dublin, Ireland, **9** Mathematics, School of Science, RMIT University, Melbourne, Australia, **10** Faculty of Life Sciences, Tel Aviv University, Tel Aviv, Israel

* t.parisini@imperial.ac.uk (TP); r.shorten@imperial.ac.uk (RS); lewistone100@gmail.com (LS)

**Data Availability Statement:** All relevant data are within the manuscript file.

**Funding:** R.S., T.P., P.C. & R.M.-S. acknowledge support from EPSRC project EP/V018450/1. R.M.-S. and S.S. acknowledge funding support from

## Abstract

COVID-19 abatement strategies have risks and uncertainties which could lead to repeating waves of infection. We show—as proof of concept grounded on rigorous mathematical evidence—that periodic, high-frequency alternation of into, and out-of, lockdown effectively mitigates second-wave effects, while allowing continued, albeit reduced, economic activity. Periodicity confers (i) predictability, which is essential for economic sustainability, and (ii) robustness, since lockdown periods are not activated by uncertain measurements over short time scales. In turn—while not eliminating the virus—this fast switching policy is sustainable over time, and it mitigates the infection until a vaccine or treatment becomes available, while alleviating the social costs associated with long lockdowns. Typically, the policy might be in the form of 1-day of work followed by 6-days of lockdown every week (or perhaps 2 days working, 5 days off) and it can be modified at a slow-rate based on measurements filtered over longer time scales. Our results highlight the potential efficacy of high frequency switching interventions in post lockdown mitigation. All code is available on Github at https://github.com/V4p1d/FPSP_Covid19. A software tool has also been developed so that interested parties can explore the proof-of-concept system.

## Author summary

*Why?* The design of post-lockdown mitigation policies while vaccines are still not available is pressing now as new secondary waves of the virus have emerged in many countries (for example, in Spain, France, UK, Italy, Israel, and others), and as several of these countries grapple with the reintroduction of full lockdown measures. *What do we do and find?* We propose efficacious and realisable methods based on control theory to tame the complex behaviour of COVID-19 in well mixed populations. We achieve this through a policy

EPSRC grant EP/R018634/1, "Closed-loop Data Science". M.B. and T.P. acknowledge funding support from the European Union's Horizon 2020 Research and Innovation Programme under Grant Agreement No 739551 (KIOS CoE). H.L. and R.S. acknowledge the support of Science Foundation Ireland. P.F. acknowledges the support of IOTA Foundation (SFI grant 16/IA/4610). L.S. acknowledges support from the Australian Research Council (ARC) from Discovery Grant DP 170102303. The funders had no role in study design, data collection and analysis, decision to publish, or preparation of the manuscript.

**Competing interests:** The authors have declared that no competing interests exist.

of fast intermittent lockdown intervals with regular period. We illustrate how our approach offers a fundamentally new perspective on ways to design COVID-19 exit strategies from policies of total lockdown. Our theoretical results are also very general and apply to a wide range of epidemiological models. *What do these findings mean?* Unlike many other proposed abatement strategies, which have risks and uncertainties possibly leading to multiple waves of infection, we demonstrate that our proposed policies have the potential to suppress the virus outbreak, while at the same time allowing continued economic activity. These policies, while of practical significance, are built on rigorous theoretical results, which are to the best of our knowledge, new in mathematical epidemiology. An extensive validation is carried out using a detailed epidemic model validated on real COVID-19 data from Italy and published very recently in *Nature Medicine*.

## Introduction

The rapid spread of COVID-19 in early 2020 forced governments to move rapidly into a prolonged lockdown to curb the spread of the virus [1–3]. Many governments have already expended enormous economic resources in dealing with the societal, health, economic, and other costs of this first lockdown [4]. Furthermore, the resulting high emotional cost placed on society is likely to make future lockdowns of this type more and more difficult to realise with high levels of compliance. Hence, a major issue is whether it is possible to design new lockdown strategies, in a manner that manages the virus, that places less emotional stress on populations, and at the same time, allows significant economic activity [5–7]. Indeed, economists are currently pursuing in depth analyses of the complexities and tradeoffs of lockdowns [8].

Several kinds of COVID-19 abatement strategies are currently implemented worldwide. These include: (i) contact tracing with/without testing; (ii) social distancing; and (iii) the possible introduction in some jurisdictions of immunity passports [9]. As a complement to these measures, several governments are considering using *intermittent lockdown interventions* driven by measured clinical data (such as demand on health care facilities, for example). As a result of these interventions, new waves of COVID-19 emerge [10]; see for example the local lockdown in Leicester, England [11]. In this respect, one of the great difficulties in designing intermittent lockdown policies based on real-time measurements comes from the many sources of uncertainty that surround the COVID-19 disease. These include: the delays and uncertainties of the available measurements; the time taken for an individual to become symptomatic, to become infectious, and to recover; the infection pathways (aerosols, surfaces); the proportion and infectivity of asymptomatic individuals, and many other factors. A further source of uncertainty is related with the unknown transmission between groups in different geographic areas and/or different geographic groups. All of these uncertainties are exacerbated by the exponential growth rate of the epidemic. Together, uncertainty, delay and exponential growth, make any data-driven timing of lockdown, as well as the optimal timing of release-from lockdown, very difficult to determine (as we shall see later).

The proof-of-concept strategy addressed in this work refers to a class of robust *periodic pulsed intervention policies* to mitigate a post lockdown epidemic while allowing economic activity. These intervention policies act on the evolution of the epidemic by orchestrating society at relatively short intervals into, and out-of, lockdown with a switching period which is *independent from measurement*. It is important to note that pulsed interventions have been dealt with in several works in epidemiology. For example, there is strong empirical support for using switching mitigation strategies in other related contexts if we refer to studies on

recurrent seasonal infectious diseases such as influenza or childhood infectious diseases (measles, chickenpox, mumps) where change of seasons has been shown to induce recurrent epidemic dynamics, that are often annual in pattern. It is also worth noting that empirical and theoretical studies have confirmed that the switching theory—as a model of seasonal forcing—is appropriate, as can be found discussed and modelled in numerous epidemiological papers (see, for instance [12–14]). Indeed, periodic vaccination policies and periodic quarantines for viral epidemics are discussed in [15–17]. A notable difference between these and our work is however that we propose, and theoretically justify, switching over much shorter time scales. Very recently in the context of COVID-19 pandemic, irregular aperiodic ON-OFF triggered quarantine policies, with long lockdown periods, are proposed in the influential paper [10]. According to this study, surveillance triggers should be based on the testing of patients in critical care (intensive care units, ICUs), with quarantine triggered whenever the number of critical care hospital beds rises above a given threshold. However, we argue that this aperiodic policy is not robust as it suffers from the above-mentioned uncertainties [2] which can themselves generate instabilities and secondary waves, as will be demonstrated (see the Discussion section). In particular, we also argue that the unknown onset of lockdowns and their unknown duration, arising from such policies, make sustained economic activity difficult.

Our findings illustrate that *regular and repeated short periods of lockdown, followed by even shorter periods of normal activity (or "opening up")*, may be effective and robust in mitigating the epidemic should a second wave occur. Further, they also allow for sustained and planned economic activity. In addition, our theoretical results provide—for the first time to the best of the authors' knowledge—a solid and rigorous ground to this high-frequency pulsed intervention policy. One embodiment of this might, for example, be repeating a week where one normal workday is followed by lockdown for the next six days of the week. Importantly, these regular periodic lockdown strategies are robust with respect to uncertainty as lockdown periods are not triggered by measurements as in [10], but rather are driven by *predictable periodic time-driven triggers in- and out- of lockdown*. We support our findings by an extensive validation using a very recent COVID-19 compartmental SIR-like mass-action model (SIDARTHE) (see [18]) derived from clinical data from the most affected region in Italy (Lombardy). In this respect, it is worth remarking that the use of SIR-like models for qualitative validation of switching on and off the lockdown phase is currently widely used (see, for example the very recent paper [6] and the work [19] presenting the SEPIAHQRD model), and does indeed capture viral transmission in a well mixed population. Furthermore, models of this type are also readily extended to multiple societal and geographic compartments [20].

The theoretical results that we derive formalise the intuition that pulsing results in an *average* epidemic behaviour that is, in a sense, *between* that associated with *lockdown* and that associated with *normality*. As we shall show, judiciously choosing the policy's period and the relative number of lockdown days within each period induces an *average* behaviour that makes it possible to drive the epidemic towards the desired compromise between economic activity and epidemic growth. It is worth noting that a number of theoretical and empirical studies have found that switching may be modelled by a weighted average of the two reproductive numbers (see, for example, [13, 21–23]). Most of these studies show that the stability of the model's infection free equilibrium (and thus the appearance of an epidemic outbreak or not) is dependent on the weighted average of the reproductive numbers. Our theoretical results go far beyond this elementary analysis and show that the epidemic trajectory itself is completely governed by the weighted average. In particular, we show that switching with periods of the order of days is sufficient to force the system to behave in a manner that approximates *infinitely fast* switching, giving rise to an *average epidemic* that can be engineered in a rigorous manner. We also note that while basing triggering policies on instantaneous data is dangerous precisely

because data are very uncertain (for examples hospitalisations may lag actual number of infections in the population by several weeks), over longer periods, uncertain data can be averaged, thus revealing long-term trends, such as whether mean levels of infections are increasing or decreasing. Our results demonstrate that effective policies can be found over time by carefully using the averaged data to adjust the specific number of workdays and lockdown days, at a very slow rate, to respond to both uncertainties in the measurements, and changes in the virus dynamics while the policy frequency remains fixed. Specifically, the characteristics of this open-loop intervention policy—number of lockdown versus working days over a specified short period—are modulated at a slow-rate by an *outer supervisory control loop*. The supervisor performs this adaptation by integrating and smoothing real empirical (not generated by a prediction model) COVID-19 related measurements (hospital admissions, deaths, positive tests) gathered over a longer time periods. It can therefore be designed to be robust and not suffer from the time-delays and the uncertainties inherent in the observations.

As a final comment, we note that since originally becoming available in March 2020 [24], the idea of fast periodic switching to abate COVID-19 has also been adopted and developed by several other well known groups in theoretical biology [25–27]. With regard to these latter papers, we emphasize that the unique contributions of our work are four-fold. First, to the best of our knowledge, we were the first group to propose this strategy in [24], predating other studies of this kind. Second, we give tight theoretical results to justify and inform the design of the switching strategy. Third, a supervisory outer loop design is proposed to account for the model uncertainties that is based on rigorous control theoretic concepts. Finally, our validation simulation study provides a rigorous exploration of the proposed strategy, and presents a suite of methods that can be used by policy makers to better inform decisions in fighting COVID-19.

## Results

The work [1] is just one of many recent publications confirming that the prolonged lockdown policies put in place by different governments in eleven European countries led to a major decrease of the COVID-19 outbreak growth rate in the respective countries (see also [28–30] regarding China, Italy, Spain, and UK). However, official governmental data also tell that these lockdown periods came with severe socio-economic consequences making extended lockdown periods unsustainable (as an example, refer to p. 2 of the USA Bureau of Labor Report [31] where it is mentioned that in the USA 5.2 million people in August 2020 were prevented from looking for work due to the pandemic and also refer to p. 1 of the EU Eurostat Report [32] in which the flash estimate for the first quarter of 2020 has GDP down by 3.8% in the euro area and by 3.5% in the EU compared with the first quarter of 2019). Hence, a compromise between virus outbreak mitigation and economical growth has to be sought. Since pulsed intervention policies have been empirically shown to be effective in epidemiology (see references in the introductory remarks), our starting point for our study is the question *whether fast alternation of lockdown and normal society functioning would yield this compromise*? We demonstrate that by modifying the epidemic's dynamics towards an average behaviour, a *Fast Periodic Switching Policy* (FPSP) leads to the above-mentioned compromise.

In what follows we organise this part of the paper in several sections to illustrate our proposed policy, and our main findings. In Part (i), we present our main idea, the FPSP policy. In Part (ii) we give a description of the theoretical justification of this policy. In Parts (iii) and (iv) we discuss the design of the outer supervisory control loop. Finally in Parts (v) and (vi) we describe the SIDARTHE model and discuss the efficacy of the overall strategy.

### (i) The fast periodic switching policy (FPSP)

After an initial phase in which the virus started infecting a completely susceptible population without any constraint, a prolonged lockdown has been enforced in several countries with the goal of substantially reducing contacts between individuals, reducing transmission pathways for the virus to spread, and thereby suppressing the epidemic. The basic idea is to apply FPSP at a given time $t_0$ after such a prolonged lockdown, with the assumption that in lockdown, the dynamics of the spread of virus are such that the virus is mitigated.

The application of the FPSP consists of allowing society to function as normal for $X$ days, followed by social isolation of $Y$ days (in short, this FPSP is denoted as $[X, Y]$). This is then repeated in every subsequent time-interval $[t_k, t_{k+1})$, $k \geq 1$, at a given *constant and suitably-high frequency* $1/T$, with $T = t_{k+1} - t_k = X + Y$ denoting the period (hence the *fast periodic* nature of the switching policy).

As formally shown later, pulsing at high frequency is important because high-frequency pulsed policies make the epidemic dynamics similar to that of an *average epidemic* characterised by a growth rate which is a weighted average of that during lockdown days and that under normal functioning days. This may seem an obvious observation. However, it is not. It is well known from the theory of switched systems [33] that switching may introduce instabilities, or even chaotic behaviour, into a dynamical systems, even when switching between subsystems that are benign and well behaved. Indeed, even when the switching system is well behaved, switching may give rise to oscillatory behaviour, or chattering behaviour, depending on the nature of the switching. A remarkable result in the context of our work is that switching over time-scales of days gives rise to this latter behaviour, and that the epidemic can essentially be "designed" to die-out in a pre-specified manner, by adjusting the parameters of our switching policy [33]. More specifically, we show that an averaged epidemic can be realised as follows. The weighted average growth rate is determined by the *policy's duty-cycle* $DC = X/T = X/(X + Y)$, that is, the relative number of non-lockdown days on each period. *The approximation error, i.e. the difference between the actual epidemic dynamics under the pulsed policy and that of the average epidemic, decreases at least linearly as the policy period decreases.* In turn, our results show that higher frequencies outperform lower ones in terms of epidemics growth.

To explain how FPSPs affect the behaviour of the epidemic, we make use of the important epidemiological index, the basic reproductive number $\mathcal{R}_0$, which characterises the number of secondary infected cases produced by a typical infected person in a fully susceptible population. It is well known that the virus grows if $\mathcal{R}_0 > 1$, and an outbreak ensues [34]. It is important to also note that even when the population is not fully susceptible, a sufficient condition for disease die-out is $\mathcal{R}_0 < 1$. In our setting, we denote by $\mathcal{R}_0^+ > 1$ the basic reproductive number of the uncontrolled outbreak, and by $\mathcal{R}_0^- < 1 < \mathcal{R}_0^+$ the one induced by a permanent lockdown policy. It turns out that, by alternating sufficiently-fast lockdown days and normal work days, the FPSP $[X, Y]$ is able to modify the reproductive number of the epidemic to a given $\mathcal{R}_0^* \in (\mathcal{R}_0^-, \mathcal{R}_0^+)$, whose exact value directly depends on the duty-cycle DC. Our theoretical results demonstrate that the epidemic trajectory itself follows closely, at each time, that generated by a model with no switching, but with growth given by the weighted average $\mathcal{R}_0^*$ (the convergence to the weighted average $\mathcal{R}_0^*$ under the action of the FPSP is strictly related to the already-mentioned fact that switching may be modelled by a weighted average of the two reproductive numbers [13, 21–23]). Further, it is worth noting that our FPSP mitigation strategy is also grounded on recent experiences of the single-shot interventions that were successful in cities and countries around the world. Detailed assessments are still underway, but through lockdowns many countries were able to reduce the reproductive number to $\mathcal{R}_0 < 1$. (In [6] it is mentioned that cities in China were able to reduce their $\mathcal{R}_0$ from 2.5 down by $50\% - 85\%$

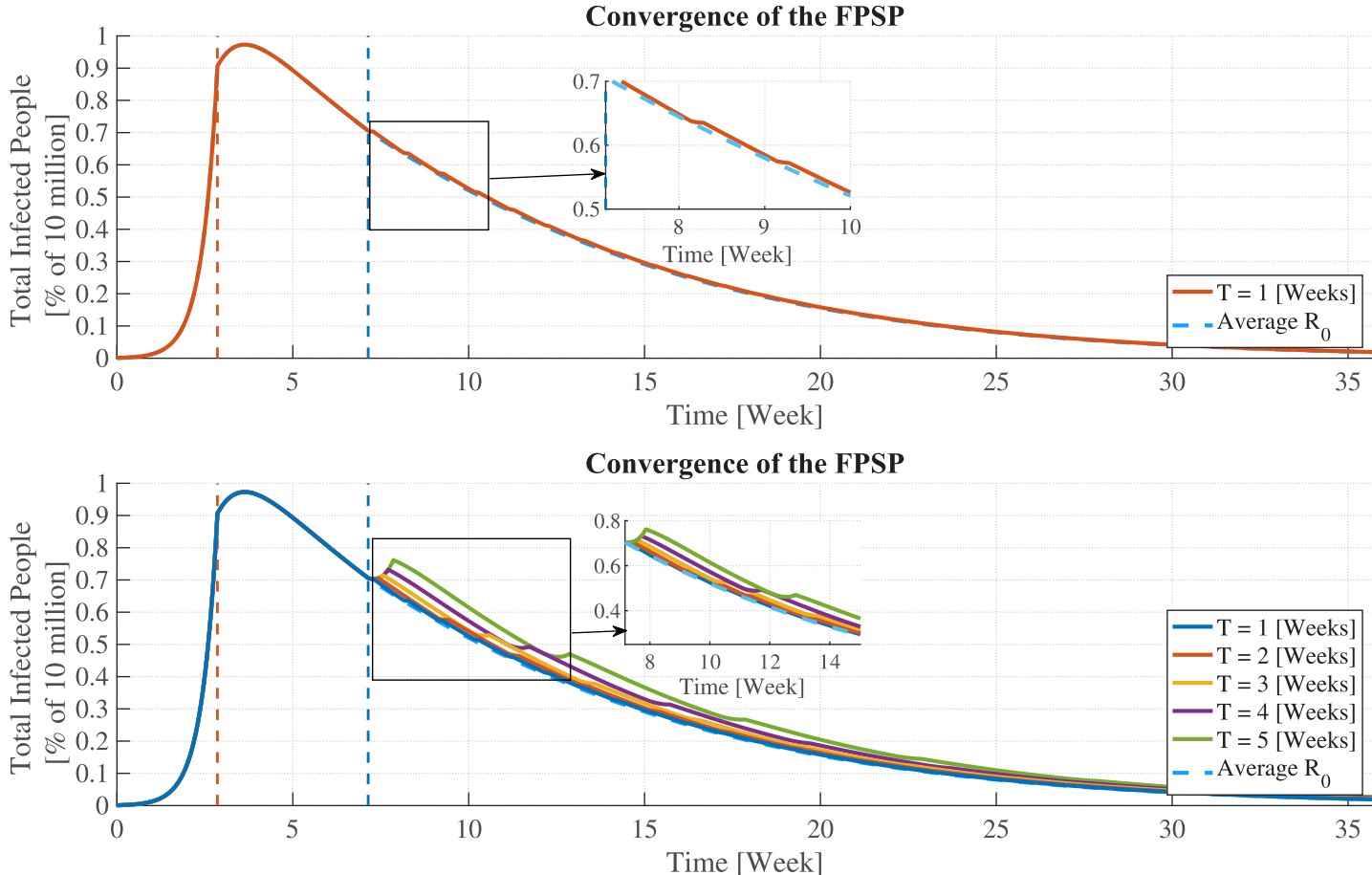

**Fig 1. Convergence of FPSP policies and $\mathcal{R}_0$.** (Top) Time-behaviour of a 7-days FPSP policy with [$X$, $Y$] = [1, 6] with a single workday followed by six days lockdown. In the first 20 days (see the vertical orange dashed line) the virus invades a totally susceptible population generating a major epidemic with reproductive number $\mathcal{R}_0^+ = 2.404$. From the end of week 3 to the end of week 6 a major lockdown is enforced reducing the reproductive number to $\mathcal{R}_0^- = 0.42$. The FPSP is initiated in week 7 (see the vertical dashed light blue line). The epidemic has a trajectory that approximates that of a system with average $\mathcal{R}_0^* = 0.734$. (Bottom) Time-behaviours generated by five FPSPs of increasing frequency (corresponding to periods $T$ ranging from 5 to 1 weeks) and the same duty-cycle DC = 1/7 = 2/14 = 3/21 = 4/28 = 5/35 $\simeq$ 14.3%. In all cases, the epidemic behaviour seen in the first three weeks is suppressed and the outbreak dies out following closely the trajectory of an un-switched system with $\mathcal{R}_0^* = 0.734$ (dashed blue line). As clearly show in the figure, smaller periods are associated with a higher vicinity to the average trajectory $\mathcal{R}_0^*$ (dashed blue line).

and several countries have now reduced their $\mathcal{R}_0 \simeq 2$ to no growth situation $\mathcal{R}_0 < 1$ as in Australia under lockdown, for instance. Indeed, for eleven European countries, Flaxman et al. [1] estimate that the average lockdown $\mathcal{R}_0$ of 0.44 [0.26-0.61] for Norway to 0.82 [0.73-0.93] for Belgium, and on average an 82% reduction compared to pre-intervention values).

Before proceeding with the theoretical foundations behind the FPSP, by way of example, we consider the application of FPSP to a COVID-19 compartmental SIR-like mass-action model (SIDARTHE) introduced in [18] and that will be extensively used throughout the paper. The time series of infected individuals for this model is plotted in Fig 1 (Left), and is produced by applying an FPSP [$X$, $Y$] = [1, 6] policy allowing in each period 1 workday followed by 6 lock-down days. In the first 20 days (see the vertical orange dashed line) the virus invades a totally susceptible population generating a major epidemic with reproductive number $\mathcal{R}_0^+ = 2.404$. From the end of week 3 to the end of week 6 a major lockdown is enforced reducing the reproductive number to $\mathcal{R}_0^- = 0.420$. The application of the FPSP is initiated in week 7 (see the vertical dashed light blue line). Then—as revealed by the trajectories in Fig 1—our argument

shows that the overall behaviour of the epidemic subject to this switching policy evolves *approximately* as an epidemic characterised by a reproductive number of $\mathcal{R}_0^* = (\mathcal{R}_0^+ + 6\mathcal{R}_0^-)/7 = 0.734 < 1$, and the approximation error decreases when the policy's frequency increases. We emphasise that without applying the FPSP, $\mathcal{R}_0$ would be greater than one in week 7, and the epidemic would grow exponentially. Instead, the FPSP ensures $\mathcal{R}_0 < 1$ and the epidemic dies out. Given the asymptotic nature of this result, it is natural to ask how well the [1, 6] policy approximates $\mathcal{R}_0^*$. The answer is depicted in Fig 1 (Bottom) showing the time-behaviours generated by five FPSPs of increasing frequency (corresponding to periods ranging from 5 to 1 weeks) and same duty-cycle DC = 1/7 = 2/14 = 3/21 = 4/28 = 5/35 $\simeq$ 14.3%: it can be observed that the evolution of epidemic approximates closer and closer the dynamics of the average system with no switching. Note, as we have already mentioned, that periods of one week give a very good approximation to the average dynamics.

## (ii) Theoretical underpinnings of FPSP

We now turn to providing the theoretical ground for FPSPs in terms of their frequency $1/T$ and duty-cycle DC in the context of SIR-like mass-action models. The dynamics of a general family of compartmental SIR-like models can be described by a differential equation of the form

$$\frac{dx(t)}{dt} = f_0[x(t)] + \sum_{i=1}^{m} \beta_i(t) f_i[x(t)], \tag{1}$$

in which $n$ denotes the dimension of the state space, $x(t) \in \Re^n$ denotes the state vector, $f_i: \Re^n \to \Re^n$, $i = 0, \ldots, m$, are continuously differentiable functions, and $\beta_i$, $i = 0, \ldots, m$, are essentially bounded functions with dimensionless values in [0, 1] that modulate the transfer rates between compartments. In this respect, it is worth noting that Eq (1) can be used to describe compartmental models with $n$ compartments in which each state component $x_i(t)$ corresponds to a different compartment. The value of each $x_i(t)$ may represent either the number of individuals or the percentage of individuals in the $i$-th compartment. In the first case, the physical dimension of $x_i(t)$ is individuals, in the second case $x_i(t)$ is dimensionless. In passing from a description to the other the functions $f_k$, $k = 0, \ldots, m$ need to be rescaled. In particular, if the functions $f_k$ describe the dynamics of the number of individuals in each compartments, the corresponding dynamics describing the percentage of individuals is given by the functions $f_k'(\cdot) := f_k(N\cdot)/N$. Similarly, passing from percentages to numbers of individuals requires the inverse scaling $f_k(\cdot) = Nf_k'(\cdot/N)$.

Furthermore, the functions $\beta_i$ will be used to model the effect of a lockdown on the dynamics of Eq (1). If $\beta_i(t) = 1$, then the lockdown is not enforced and the transfer rates are maximal. If $\beta_i(t) < 1$, a lockdown is enforced, and the transfer rates between any two compartments are reduced accordingly. For example, a basic SIR model describing the spread of a virus with time-varying basic/effective reproductive number in a population of $N$ individuals can be written in the form of Eq (1), with $n = 3$, $x = (S, I, R)$, in which $S(t), I(t), R(t) \in [0, N]$ denote, respectively, the number of susceptible, infected, and recovered individuals, $m = 1$ and

$$f_0(x) \quad := \begin{pmatrix} 0 \\ -\alpha I \\ \alpha I \end{pmatrix}, \qquad f_1(x) \quad := \frac{1}{N} \begin{pmatrix} -\sigma SI \\ \sigma SI \\ 0 \end{pmatrix} \tag{2}$$

for some parameters $\alpha, \sigma > 0$ modelling, respectively, the recovery rate and the rate of effective contacts between infected and susceptible individuals. Rates are in 1/day. Here, $\beta_1(t)$

modulates the infection rate $\sigma$ and, thus, the basic/effective reproductive number. In a similar way, many existing more comprehensive SIR-like models have dynamics described by Eq (1) for a suitable choice of the state vector, the integers $n$ and $m$, and the functions $f_i$, $i = 1, \ldots, m$. By imposing lockdown followed by work days, the application of a FPSP policy to an epidemic described by Eq (1) makes each function $\beta_i$ to switch between two different values. For simplicity, we consider the case in which $\beta_i^+$ and $\beta_i^-$ are both constant. We remark, however, that this assumption does not affect the qualitative claims of the presented theory, and that this comes without loss of generality, since $\beta_i^+$ and $\beta_i^-$ can be taken as worst-case values. Specifically, we can write

$$\beta_i^{\text{FPSP}}(t) = \begin{cases} \beta_i^+ & \text{during non lockdown days (society functioning as normal)} \\ \beta_i^- & \text{during lockdown and social isolation}, \end{cases} \tag{3}$$

for some $\beta_i^+, \beta_i^- \in [0, 1]$ satisfying $\beta_i^+ \geq \beta_i^-$. Typically, $\beta_i^+ = 1$, which corresponds to an unmitigated infection rate. If other mitigation measures are in place, such as mandatory use of face masks or social distancing policies, then $\beta_i^+$ may be smaller than 1. The switching period $T$ and the duty-cycle DC are defined by the particular pair $[X, Y]$ identifying the particular FPSP policy (recall that $T = X + Y$ and DC $= X/T$). For each $i = 1, \ldots, m$, we define the *average mode* $\beta_i^*$ as the weighted average between the two modes $\beta_i^+$ and $\beta_i^-$, namely:

$$\beta_i^* := DC \cdot \beta_i^+ + (1 - DC) \cdot \beta_i^-.$$

The value of $\beta_i^*$ equals the *weighted mean value* of $\beta_i$ over each period interval of time of length $T$. As clear from the definition, $\beta_i^*$ may coincide with any value in the interval $[\beta_i^-, \beta_i^+]$ for a suitable duty-cycle DC. Letting $\beta^* = (\beta_1^*, \ldots, \beta_m^*)$ and referring to the differential model (1) obtained with $\beta(t) = \beta^*$, the *average dynamics* is characterised by the model

$$\frac{dx(t)}{dt} = f_0[x(t)] + \sum_{i=1}^{m} \beta_i^* f_i[x(t)]. \tag{4}$$

For a given FPSP policy characterized by the pair $[X, Y]$, and with $\beta_i^{\text{FPSP}}$ defined by Eq (3) for each $i = 1, \ldots, m$, denote by $x^{\text{FPSP}}$ the unique solution to Eq (1) originating at $x_0^{\text{FPSP}} := x^{\text{FPSP}}(0)$ at time $t = 0$, and obtained through $\beta_i(t) = \beta_i^{\text{FPSP}}(t)$ for all $i = 1, \ldots, m$ which is generated by applying the FPSP policy. Likewise, denote by $x^*$ the unique solution to Eq (4) originating at $x_0^* := x^*(0)$ at time $t = 0$. Then, we show that (see Theorem 1 and its proof in the Methods section), for every time instant $\tau \geq 0$, there exist $a_1, a_2 \geq 0$, such that the following bound holds:

$$\|x^{\text{FPSP}}(t) - x^*(t)\| \leq a_1 \|x_0^{\text{FPSP}} - x_0^*\| + a_2 T, \qquad \forall t \in [0, \tau]. \tag{5}$$

Inequality (5) has several implications discussed hereafter. First, if $x_0^{\text{FPSP}} = x_0^*$ (i.e., solutions starting from the same initial conditions are considered), then Inequality (5) reduces to

$$\|x^{\text{FPSP}}(t) - x^*(t)\| \leq a_2 T, \qquad \forall t \in [0, \tau],$$

which formally shows that *large frequencies lead to smaller approximation errors with respect to the average dynamics*. This, in turn, is a fundamental finding which conveniently separate the FPSP policies from existing measurement-based policies. Indeed, in the latter approaches the need to cope with delays and uncertainties necessarily lead to quite small switching frequencies, while we show here that the frequency should be taken as large as possible. Moreover, by properly selecting a sufficiently small period $T$ and suitable values of the duty-cycle DC, the

FPSP is able to *shape* the dynamics of the epidemic (for instance, by designing parameters $\beta_i^*$ in such a way that $\mathcal{R}_0 < 1$) as confirmed by the example shown in Fig 1 and the related discussion. Since $\mathcal{R}_0^- < 1$ and DC can take any value in [0, 1], the latter observation implies in particular that we can always find a suitable value DC of the duty-cycle for which the average dynamics described by Eq (4) is monotonically decaying, and a sufficiently small period $T$ for which the actual epidemic dynamics induced by a $T$-periodic FPSP policy is arbitrarily close to such stable average evolution. The precise value of the duty-cycle and period yielding this behaviour, however, are strongly model and parameter dependent, and they are not necessarily feasible for implementation. Nevertheless, our forthcoming results show that, for the considered models of the COVID-19 outbreak, reasonable values of the duty-cycle (e.g. 1 or 2 days per week) and of the period (e.g. 1, 2 or 3 weeks) succeed in producing good results.

### (iii) Slow outer supervisory feedback loop

The FPSP open-loop intervention policy is augmented by an adaptive component helping the policymaker in deciding when and how to change the policy duty-cycle. Based on clinical data averaged over suitably long time periods, this component of our system seeks to *automatically* find the periodic policy that represents a good compromise between abatement of the virus, and economic activity. Specifically, starting from a very conservative policy that is close to a full lockdown, for example 1 day of activity followed by 6 days of lockdown, the objective of this component is to gradually adjust the policy based on clinical data averaged over long periods of time. This part of the policy is called the *slow outer supervisory control loop* (in the following named or "outer loop" for simplicity) that selects at each time $t_k$, $k \geq 1$ the specific pair $[X(t_k), Y(t_k)]$ (where recall $X(t_k) + Y(t_k) = T$ and hence the duty-cycle DC $= X(t_k)/T$)) on the basis of the observed levels of rate of infection over longer timescales. We stress that the outer supervisory control loop does not change the period of the FPSP, but only its duty-cycle. Thus, the outer loop can make decisions slowly enough to handle the delays in the measurement, without constraining the FPSP switching frequency. The outer loop is designed as an hysteresis-based control scheme that is characterised by simplicity of implementation and by its *inherent robustness*. Specifically, we let $t_0$ be the time-instant when the control action starts (i.e., the end of a prolonged lockdown) and we set $X(t_0) = 0$, $Y(t_0) = T$. Then, by considering the half-closed intervals $[t_k, t_{k+1})$, with $t_{k+1} - t_k = T$, the hysteresis-based supervisory outer control law can be expressed as follows:

$$X(t_{k+1}) = \begin{cases} \mathrm{mid}(0, X(t_k) + 1, T), & \text{if } \psi_X(t_{k+1}) > 0, \\ \mathrm{mid}(0, X(t_k) - 1, T), & \text{if } \psi_Y(t_{k+1}) > 0, \\ \mathrm{mid}(0, X(t_k), T), & \text{otherwise} \end{cases} \tag{6}$$

$$Y(t_{k+1}) = T - X(t_{k+1}), \tag{7}$$

where

$$\mathrm{mid}(a, b, c) = \begin{cases} a, & \text{if } b \leq a \\ b, & \text{if } a < b < c \\ c, & \text{otherwise.} \end{cases}$$

In Eqs (6) and (7), functions $\psi_X$ and $\psi_Y$ are given by

$$\psi_X(t_{k+1}) = (1 - \alpha_X) \int_{t_{k-1}-\Delta}^{t_k-\Delta} [O(s) - O(t_{k-1} - \Delta)]ds - \int_{t_k-\Delta}^{t_{k+1}-\Delta} [O(s) - O(t_k - \Delta)]ds,$$

$$\psi_Y(t_{k+1}) = -(1 + \alpha_Y) \int_{t_{k-1}-\Delta}^{t_k-\Delta} [O(s) - O(t_{k-1} - \Delta)]ds + \int_{t_k-\Delta}^{t_{k+1}-\Delta} [O(s) - O(t_k - \Delta)]ds.$$

where $\alpha_X$, $\alpha_Y$ represent two positive design constants, $O(t)$ denotes the observed amount of infected people (or other meaningful measurement such as deaths or ICUs), and $\Delta < T$ is the expected delay (related to the expected incubation period) affecting the measurement of the amount on infected people. Notice that the values $\psi_X(t_{k+1})$ and $\psi_Y(t_{k+1})$ are the ones that are used by the outer loop to determine a variation on $X(t_{k+1})$ and $Y(t_{k+1})$ and they both depend on the integral of the values of $O(t)$ over the time interval $(t_{k-1}-\Delta, t_{k+1} - \Delta)$ (in a realistic scenario they would be approximated by the sum of the daily reports during the time interval $[t_{k-1} - \Delta, t_{k+1} - \Delta)$). It is also worth noting that the delay $\Delta$ in the observed number of infectives is explicitly taken into account in the design of the outer loop (hence the effect of the delays are filtered out). As a final observation on the outer controller, we want to stress that the choice of the initial conditions ($X(t_0) = 0$, $Y(t_0) = T$) and the increases to $X(t_k)$ in Eq (6) could be in theory changed, given more information on the disease. However, due to the critical nature of the epidemic, the authors believe that a "gentle" approach that applies changes to the FPSP in a gradual way represents the more advisable option.

## (iv) Overview of outer loop properties

The basic design principle justifying the outer loop (6) and (7) is very well-known in the control community under the name of "hysteresis" or "thermostat" based control (see, for instance, [35, 36]). In fact, it is a well-established control scheme that, due to its implementation simplicity and robustness to delays and uncertainties, boasts numerous industrial applications (see, for instance, [36, 37] and the references therein). In this specific context, under the assumption that the effect of each duty-cycle on the growth rate of the measurements $O(t)$ is stationary for a long enough time—meaning that a given duty-cycle has the same effect on the measured signal in a large enough time-span—one can show that the sequence $\{(X(t_k), Y(t_k))\}_{k\in\mathbb{N}}$ of pairs produced by the outer loop (6) and (7) enjoys the following properties:

P1: If the set $\mathcal{A}$ of pairs $(X, Y)$ for which both inequalities

$$\psi_X(t) \leq 0, \qquad \psi_Y(t) \leq 0, \qquad \forall t \geq 0$$

hold is non-empty, then $\{(X(t_k), Y(t_k))\}_{k\in\mathbb{N}}$ converges, in a fixed-time bounded by $T$, to $\mathcal{A}$.

P2: If $\mathcal{A}$ is empty—meaning that each possible duty-cycle is either stabilising or destabilising—then $\{(X(t_k), Y(t_k))\}_{k\in\mathbb{N}}$ may instead oscillate between stabilising and destabilising pairs.

By suitably tuning the parameters $\alpha_X$ and $\alpha_Y$, one can adjust the average steady-state effect of the sequence $\{(X(t_k), Y(t_k))\}_{k\in\mathbb{N}}$ to ensure that, in both the aforementioned cases, the net effect of the steady-state outer loop decision is *always stabilising*. Moreover, one may always increase the number of possible duty-cycles (if necessary by increasing $T$) to guarantee that $\mathcal{A} \neq \emptyset$. Finally, it is worth observing that, under the assumption that the infected individuals always develop immunity, the measurement signal $O(t)$ eventually decreases to zero.

Therefore, the same holds for $\psi_X(t)$ and $\psi_Y(t)$. In view of Eq (6), this in turn implies that the sequence of duty-cycles produced by the outer loop always stops to an equilibrium value.

## (v) The SIDARTHE class of models

The intuition underpinning the proposed control methodology stems from a thorough analysis of the nonlinear dynamics of SIR-type dynamic models of epidemics for well mixed populations. While we have tested our FPSP approach on a portfolio of related models (deterministic and stochastic SIQR/SEIR, as well as agent based models) our principal tool of validation reported here is the SIDARTHE model. By way of background, the SIDARTHE model was developed in response to the COVID-19 outbreak in Lombardy in early Spring 2020, and all our parameter values are based on those identified in the context of this work. It thus represents a state-of-the art model of the spread on COVID-19. Specifically, general SIDARTHE SIR-like mass-action model [18] dynamics is described by the following state equations:

$$\frac{dS(t)}{dt} = -\frac{\beta(t)S}{N} \cdot (\sigma_1 I + \sigma_2 D + \sigma_3 A + \sigma_4 R)$$

$$\frac{dI(t)}{dt} = \frac{\beta(t)S}{N} \cdot (\sigma_1 I + \sigma_2 D + \sigma_3 A + \sigma_4 R) - (\sigma_5 + \sigma_6 + \sigma_7)I$$

$$\frac{dD(t)}{dt} = \sigma_5 I - (\sigma_8 + \sigma_9)D$$

$$\frac{dA(t)}{dt} = \sigma_6 I - (\sigma_{10} + \sigma_{11} + \sigma_{12})A$$

$$\frac{dR(t)}{dt} = \sigma_8 D + \sigma_{10} A - (\sigma_{13} + \sigma_{14})R$$

$$\frac{dT(t)}{dt} = \sigma_{11} A + \sigma_{13} R - (\sigma_{15} + \sigma_{16})T$$

$$\frac{dH(t)}{dt} = \sigma_7 I + \sigma_9 D + \sigma_{12} A + \sigma_{14} R + \sigma_{15} T$$

$$\frac{dE(t)}{dt} = \sigma_{16} T$$

(8)

where the state $S$ represents the *susceptible* population; $I$ represents the asymptomatic, undetected *infected* population; $D$ represents the *diagnosed* people, corresponding to asymptomatic detected cases; $A$ represents the *ailing* people, corresponding to the symptomatic undetected cases; $R$ represents the *recognized* people, corresponding to the symptomatic detected cases; $T$ represents the *threatened* people, corresponding to the detected cases with life-threatening symptoms; $H$ represents the *healed* people; $E$ represents the *extinct* population. Moreover, the parameters $\sigma_1$, $\sigma_2$, $\sigma_3$ and $\sigma_4$ denote the transmission rates from the susceptible state to any of

the four other infected states; the parameters $\sigma_5$ and $\sigma_{10}$ denote the rates of detection of asymptomatic and mildly symptomatic cases; the parameters $\sigma_6$ and $\sigma_8$ denote the rates by which (asymptomatic and symptomatic) infected subjects develop clinically relevant symptoms; the parameters $\sigma_{11}$ and $\sigma_{13}$ denote the rates with which (undetected and detected) infected subjects develop life-threatening symptoms; the parameter $\sigma_{16}$ denotes the mortality rates for people who have already developed life-threatening symptoms; the parameters $\sigma_7$, $\sigma_9$, $\sigma_{12}$, $\sigma_{14}$ and $\sigma_{15}$ denote the rates of recovery for the five classes of infected subjects (including those in life-threatening conditions). All the rates are in 1/day. Finally, $N$ denotes the total size of the population and, as in Eq (1), $\beta$ modulates the rate of effective contacts between infected and susceptible individuals. For constant values of $\beta$, in this case of the SIDARTHE the basic reproduction number $\mathcal{R}_0$ is defined as follows (see [18]):

$$\mathcal{R}_0 = \beta \left( \frac{\sigma_1}{r_1} + \frac{\sigma_2 \sigma_5}{r_1 r_2} + \frac{\sigma_3 \sigma_6}{r_1 r_3} + \frac{\sigma_4 \sigma_5 \sigma_8}{r_1 r_2 r_4} + \frac{\sigma_4 \sigma_6 \sigma_{10}}{r_1 r_3 r_4} \right),$$

in which $r_1 = \sigma_5 + \sigma_6 + \sigma_7$, $r_2 = \sigma_8 + \sigma_9$, $r_3 = \sigma_{10} + \sigma_{11} + \sigma_{12}$ and $r_4 = \sigma_{13} + \sigma_{14}$. Also, it is worth noting that the model (8) can be cast into the general state Eq (1) by letting $n = 8$, $m = 1$, $x = (S, I, D, A, R, T, H, E)$, $\beta_1(t) = \beta(t)$, and

$$f_0(x) = \begin{pmatrix} 0 \\ -(\sigma_5 + \sigma_6 + \sigma_7)x_2 \\ \sigma_5 x_2 - (\sigma_8 + \sigma_9)x_3 \\ \sigma_6 x_2 - (\sigma_{10} + \sigma_{11} + \sigma_{12})x_4 \\ \sigma_8 x_3 + \sigma_{10}x_4 - (\sigma_{13} + \sigma_{14})x_5 \\ \sigma_{11}x_4 + \sigma_{13}x_5 - (\sigma_{15} + \sigma_{16})x_6 \\ \sigma_7 x_2 + \sigma_9 x_3 + \sigma_{12}x_4 + \sigma_{14}x_5 + \sigma_{15}x_6 \\ \sigma_{16}x_6 \end{pmatrix}, \tag{9}$$

$$f_1(x) = \frac{1}{N} \begin{pmatrix} -\sigma_1 x_1 x_2 - \sigma_2 x_1 x_3 - \sigma_3 x_1 x_4 - \sigma_4 x_1 x_5 \\ \sigma_1 x_1 x_2 + \sigma_2 x_1 x_3 + \sigma_3 x_1 x_4 + \sigma_4 x_1 x_5 \\ 0 \\ 0 \\ 0 \\ 0 \\ 0 \\ 0 \end{pmatrix}. \tag{10}$$

The numerical values of the parameters are identified by [18] in the context of the COVID-19 outbreak in Lombardy in early Spring 2020: $\sigma_1 = 0.570$, $\sigma_2 = 0.011$, $\sigma_3 = 0.456$, $\sigma_4 = 0.011$, $\sigma_5$

$= 0.171$, $\sigma_6 = 0.125$, $\sigma_7 = 0.034$, $\sigma_8 = 0.125$, $\sigma_9 = 0.034$, $\sigma_{10} = 0.371$, $\sigma_{11} = 0.012$, $\sigma_{12} = 0.017$, $\sigma_{13} = 0.027$, $\sigma_{14} = 0.017$, $\sigma_{15} = 0.017$, and $\sigma_{16} = 0.003$. Again, rates have dimension 1/day.

### (vi) Efficacy of FPSP and slow outer feedback loop

To begin with, in the validation simulations considered here we note that the function $\beta$ takes the values $\beta^- = 0.175$ or $\beta^+ = 1$, respectively, in case a lockdown is enforced or not. The value $\beta^+ = 1$ represents the case in which no mitigation measure is in place when lockdown is not enforced, while the value $\beta^- = 0.175$ is chosen according to [18]. Finally, the total population is set to $N = 10^7$ and the initially infected population is set to approximately 0.1% of $N$. We use $N = 10^7$ to be consistent with [18] in which the above parameters are identified. Nevertheless we remark that, being Model (8) a "mean-field" model, the population size is not important for the qualitative behaviour of the simulation.

Three subsequent simulation phases are considered: (i) for $t < 20$ days, the virus spreads with no containment measures and in this phase $\beta(t) = \beta^+ = 1$; (ii) for $20 \leq t < 50$ days, a strict lockdown is enforced and in this phase $\beta(t) = \beta^- = 0.175$; (iii) for $t \geq 50$ days (on $t = 50$ days the number of infected people has sufficiently decreased) our FPSP is activated, and $\beta(t)$ oscillates between $\beta^-$ and $\beta^+$ according to the chosen duty-cycle.

The results shown in Figs 2, 3, 4 and 5 reveal the effectiveness of the FPSP strategy and are in accordance with the theoretical results reported above. Fig 2 indicates that some switching policies are highly effective, whereas others are not. For example, in all scenarios, policies with a 29% duty-cycle (the [2, 5] policy in the case of a one week period) out-perform the 43% policy (the [3, 4] policy in the case of a one week period). In fact, provided that the period $T$ is sufficiently small, the infection decays as long as the weighted average $\mathcal{R}_0^*$ remains below 1 thus making the policy a success. In contrast, for $\mathcal{R}_0^* > 1$ the infection initially grows exponentially and the policy fails.

In the right-hand panels of Fig 2, each panel shows the epidemic dynamics for different periods $T$, from $T = 2$ weeks to $T = 4$ weeks. We see that for a fixed value of the duty-cycle, increasing the period $T$ yields a larger growth of the infection, although the time-series remain qualitatively similar.

Starting from the same above-mentioned initial condition (approximately 0.1% of the total population $N = 10^7$), we offer Figs 3 and 4 which provide a comprehensive analysis of the effect of a wide spectrum of open-loop FPSP policies in terms of the induced value and time-location of the infection peak. Specifically, Fig 3 deals of a selection of a few significant choices of duty-cycles and periods aiming at giving a qualitative understanding of the main trends of the infection as function of the chosen duty-cycle and period of the FPSP policy. Moreover, a few time-behaviours of the epidemic are shown. Fig 4 provides the full-picture of this analysis for a very large selection of different duty-cycles and periods. In particular, with reference to Fig 4, in the top-left panel for each policy we show the maximum number of infected people (i.e., the maximum value of the total infected population $I(t) + D(t) + A(t) + R(t) + T(t)$) attained after the preliminary lockdown phase is released and the FPSP policy is enforced (i.e. after $t = 50$ days). This is plotted as a function of the period $T$ of the policy used. Recall that for successful policies, decreasing $T$ should improve the approximation we make use of. In the bottom-left panel, instead, we show the time at which such infection peak is attained. The "stable" policies, which are those inducing a monotonically decreasing average infection trajectory after $t = 50$ days, attain their peak close to the starting time $t = 50$ days, and a peak value close to the starting infection value. The "unstable" policies, which are those obtained for large duty-cycles or large periods and inducing no mitigation effect, attain the infection peak quite early, and induce a very large peak value. The "middle" policies, which are those lying in-between the stable and

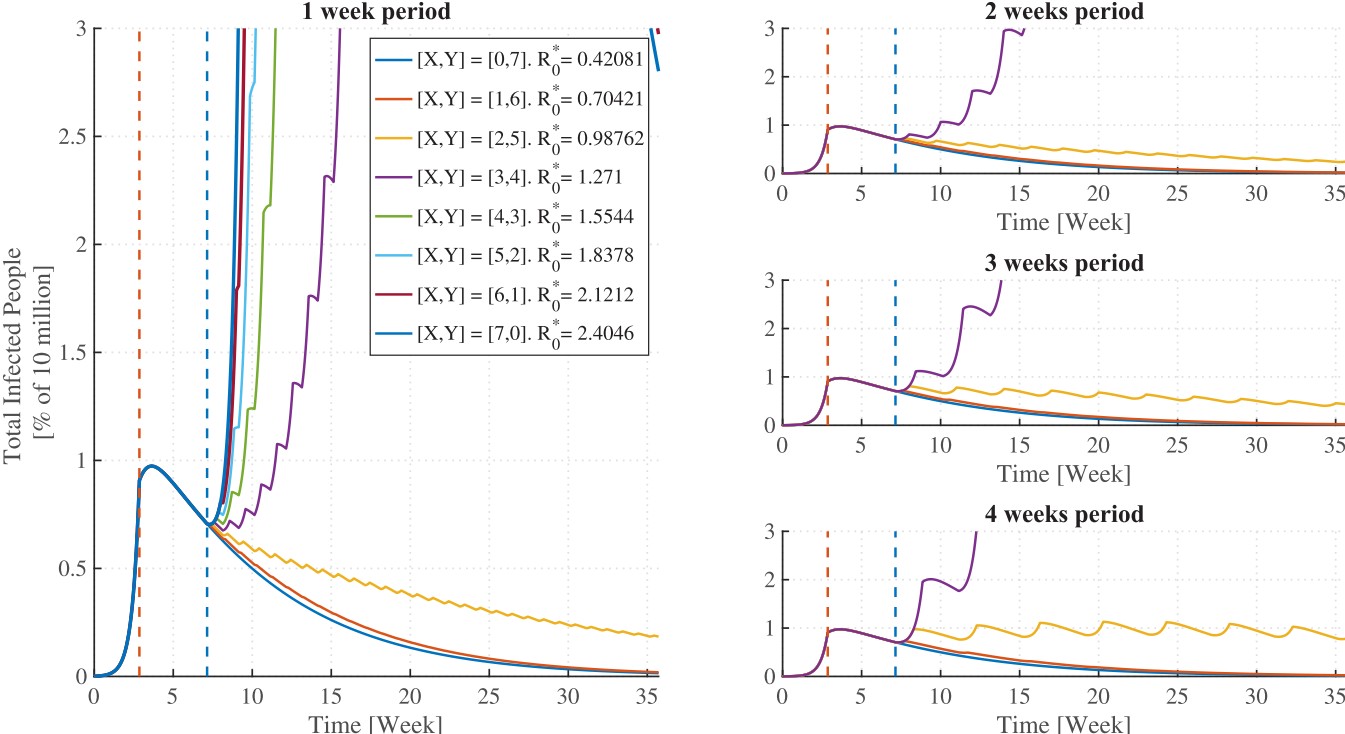

**Fig 2. The epidemic behaviour as a function of FPSP duty-cycle and period.** The total number of infected individuals plotted as a function of time for different FPSP policies. Colours distinguish duty-cycles. **(Left)** The success of the switching policy depends on whether or not the duty-cycle ensures $\mathcal{R}_0^* < 1$, in which case the epidemic dies out. Smaller duty-cycles outperform larger ones in terms of virus growth, but they lead to longer lockdowns. **(Right)** Epidemic evolution for different periods, from $T = 2$weeks to $T = 4$weeks. Across the panels, the dynamics are qualitatively the same, but they show that shorter periods perform better. For comparison, the time-behaviour corresponding to a full lockdown is also reported (FPSP [0, 7]).

unstable ones, show instead a mixed behaviour with possibly large peak-times. Again, a wide variation in the performance of policies can be observed. In particular, for similar values of the duty-cycle, which are associated with similar values of $\mathcal{R}_0^*$, we can see that *larger periods are associated with an exponential decrease of performance*. As mentioned earlier, this is consistent with the presented results, and in particular with the bound provided by Eq (5), and it reflects the fact that larger periods yield coarser approximations of the averaged trajectory $x^*$. Therefore, only for small enough period $T$, the reproductive number $\mathcal{R}_0^*$ gives a good indication of the actual behaviour of the epidemic under the action of the FPSP policy.

Fig 5 depicts the efficacy of augmentation of the open-loop policies with a slow outer loop. Here, convergence to a good policy can be clearly observed. Notice that we consider time periods of at least two weeks in order to take into account that the incubation period for the disease can be as large as 14 days. However, notice also that the update involves exclusively the duty-cycle of the FPSP and nothing prevents, for instance, to employ instead of a [3, 11] policy, a [2, 5] and a [1, 6] policy over the course of two weeks. Fig 5 also shows that lower frequencies seem to lead to lower numbers of infected people: this is only an apparent contradiction with the previous findings. Lower frequencies are in fact characterized by much larger convergence times (e.g., approximately 10 weeks for $T = 14$ against 400 for $T = 28$) which, in turn, lead to a lower duty-cycle value for longer periods than higher frequencies.

To summarise, Figs 2, 4 and 5 deliver two broad messages. First, in accordance with our theory, higher frequencies clearly outperform lower ones in terms of peak-value and peak-

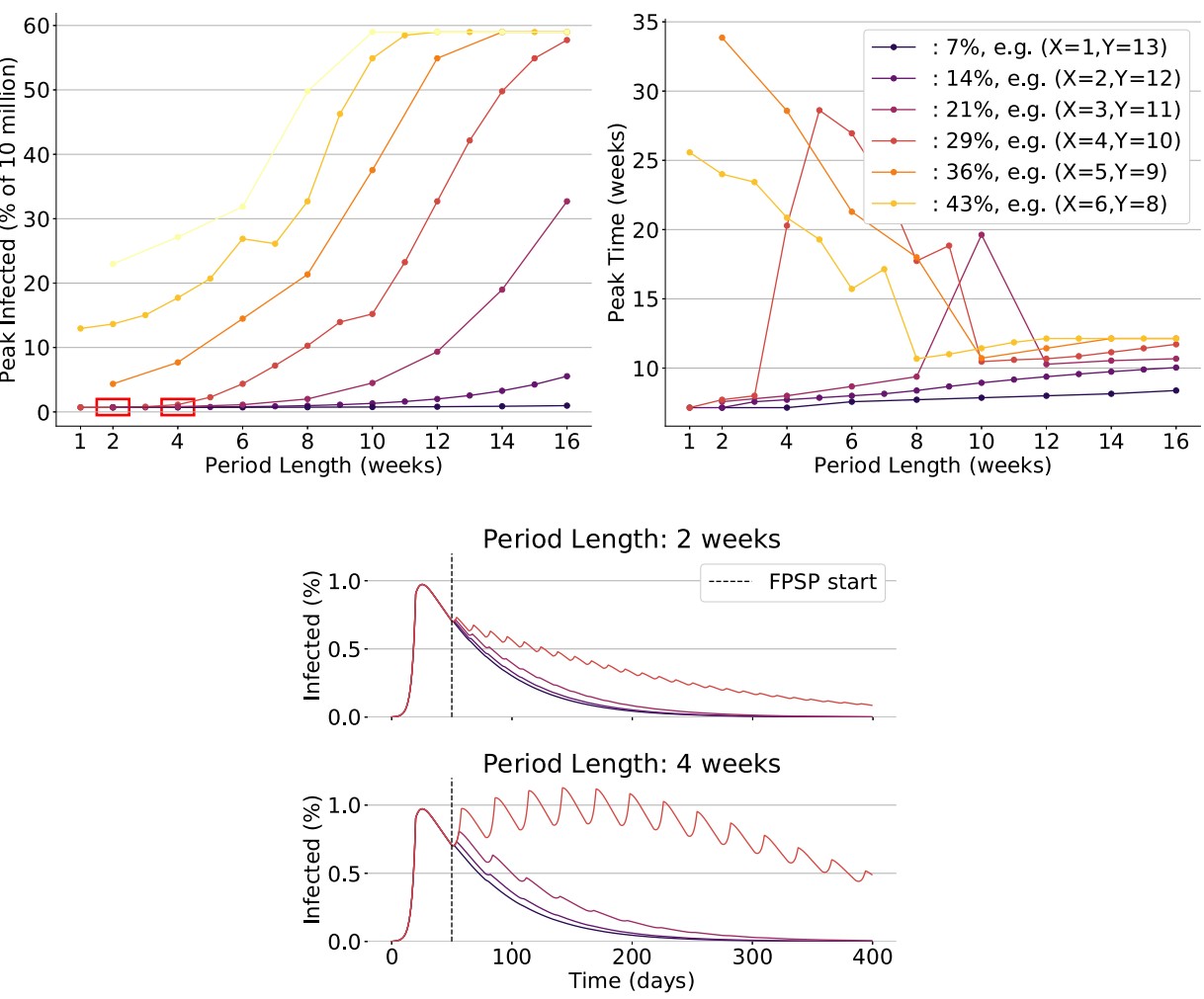

**Fig 3. Some instances of analysis of infection over time as a function of FPSP duty-cycle and associated modes of behaviours of the epidemic.** In all diagrams showing peak-values and peak-times, each point corresponds to a single policy and all policies have a period which is a multiple of seven days. Colours distinguish duty-cycles as indicated in the top-right panel. **(Top-Left)** Shows that infected peak-values increase with duty-cycle for fixed period lengths and, notably, with increased period length for fixed duty-cycles. **(Top-right)** shows that peak-times are small for policies attaining small and large peak values, while they are inversely related to the peak values for policies attaining middle peak values. **(Bottom)** highlights the time-behaviour of the epidemic for two choices of the period and four choices of the duty-cycle. These time-trajectories correspond to policies belonging to the two red rectangles shown at the bottom-left of the top-left panel.

time of infection. In addition, the outer loop always converges to a feasible and easy-to-implement regular [X, Y] policy.

We conclude the presentation of our results by illustrating in Fig 6 a schematic view of the overall mitigation strategy we suggest. First, a period $T$ is fixed once for all. According to our results and the above discussion, $T$ has to be taken as small as possible compatibly with societal constraints. Then, an initial duty-cycle DC is chosen. The pair $(T, DC)$ defines a FPSP with $X = DC \cdot T$ and $Y = T - X$. Based on empirical measurements, the supervisory controller adapts the duty-cycle—by keeping $T$ fixed—at run time to automatically find the *largest possible* duty-cycle keeping the epidemics under control.

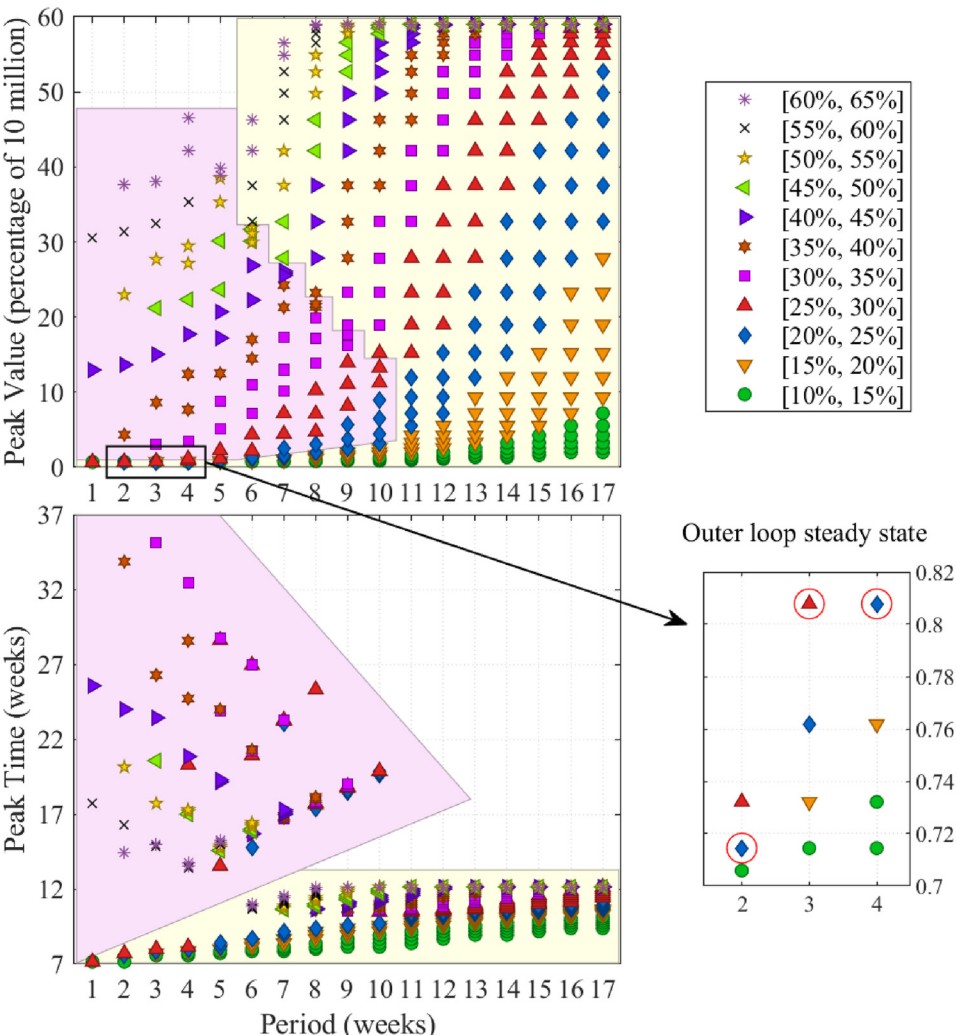

**Fig 4. Full in depth analysis of infection over time as a function of FPSP duty-cycle and period and outer loop driven equilibria.** Analogously to the instances shown in Fig 3, in all diagrams showing peak-values and peak-times, each point corresponds to a single policy and all policies have a period which is a multiple of seven days. Markers/ colours distinguish duty-cycles (e.g., blue diamonds in bottom diagrams denote policies with a duty-cycle between 20% and 25%) as seen in the top-right panel. **(Top-Left)** Shows that infected peak-values increase with duty-cycle for fixed period lengths and, notably, with increased period length for fixed duty-cycles. **(Bottom-Left)** shows that peak-times are small for policies attaining small and large peak values, while they are inversely related to the peak values for policies attaining middle peak values. Two distinct groups of policies, clustered on the basis of their peak-time behaviour, are highlighted in matching coloured regions in the **(Top-Left)** and **(Bottom-Left)** diagrams. **(Bottom-right)** highlights the duty-cycles to which the outer loop converges. Notably, these policies lie in the region of smallest peak-value (also, see Fig 5).

## Materials and methods

### Mathematical results

This section presents the basic mathematical result on which our design procedure is grounded and that have been illustrated in the **Results** section. The following notation is used: we denote by $\Re$ and $\mathcal{N}$ the set of real and natural numbers respectively, and we let $\Re_{\geq 0} := [0, \infty)$. The Euclidean norm on $\Re^n$, $n \in \mathcal{N}$, is denoted by $\|\cdot\|$. With $A, B \subset \Re$, $L^\infty(A, B)$ denotes

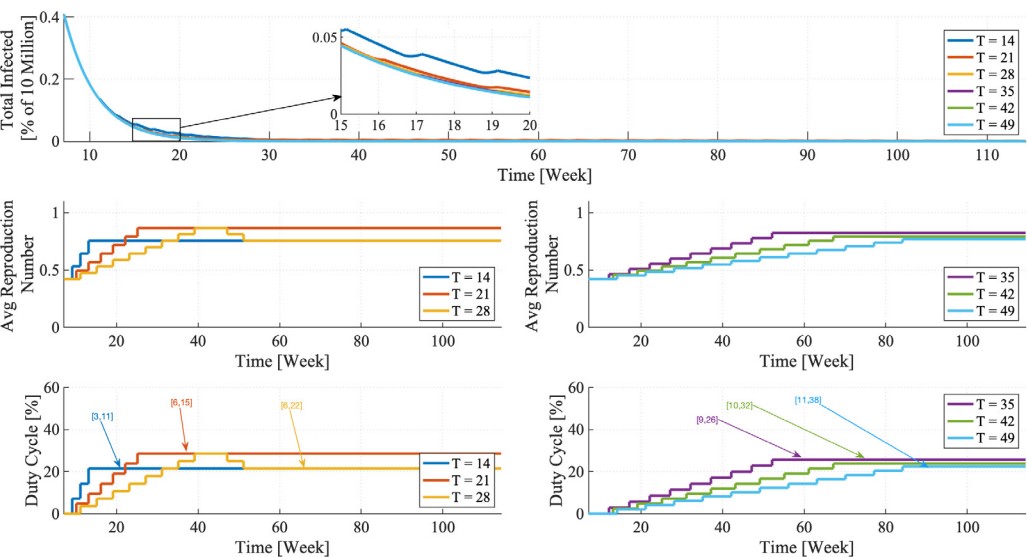

**Fig 5. Time behaviour of the outer supervisory control loop for different FPSP periods.** The periods range from 14 to 49 days. During non-lockdown, the epidemic evolves with parameters corresponding to a reproduction number $\mathcal{R}_0 = 2.38$. **(Top)** total amount of infected population. **(Middle)** Average reproductive number obtained with the duty-cycles shown in the lower panels. It can be noticed that, for each choice of the period $T$, the outer loop converges to a pair $[X, Y]$ (whose exact values are highlighted with arrows of the corresponding colour in the two lower panels) that successfully suppresses the virus (see the convergence points in Fig 4).

the Banach space of (classes of) Lebesgue measurable functions $A \to B$ which are essentially bounded, endowed with the essential supremum norm $\|f\|_\infty = \inf\{b \geq 0 \; : \; |f(x)| \leq b \text{ for almost all } x \in A\}$. With $f: A \to B$ and $g: B \to C$, $g \circ f: A \to C$ denotes the composition between $f$ and $g$. When the argument of a differentiable function $x: \Re \to \Re^n$ represents time, for simplicity its derivative $dx/dt$ is denoted by $\dot{x}$. Finally, we denote by $\lfloor \cdot \rfloor : \Re_{\geq 0} \to \mathcal{N}$ the floor function, defined as $\lfloor a \rfloor = \max\{n \in \mathcal{N} \; : \; n \leq a\}$.

With reference to the general dynamics of SIR-like models described by Eq (1), we prove the following theorem which establishes a relation between the solutions to Eq (1) obtained when the functions $\beta_i$ have the aforementioned switching behaviour, and the solutions obtained in the case in which each function $\beta_i$ is given by $\beta_i(t) = \beta_i^*$, for some constant $\beta_i^* \in [\beta_i^-, \beta_i^+]$. In particular, when each $\beta_i$ is periodic and has mean value equal to $\beta_i^*$, the result of the theorem states that the solutions obtained with the switching functions $\beta_i$ remain close to those obtained with $\beta_i(t) = \beta_i^*$, and the distance between the two decreases linearly with the switching period on each compact interval of time.

**Theorem 1** Let $\beta_i^- \leq \beta_i^+$ be arbitrary. Let $K \subset \Re^n$ be a compact set that is positively invariant for Eq (1) for every $\beta_i \in L^\infty(\Re_{\geq 0}; [\beta_i^-, \beta_i^+])$. Then, there exist strictly increasing functions $\alpha_1, \alpha_2, \alpha_3: \Re_{\geq 0} \to \Re_{\geq 0}$ such that, for each $x_0, x_0^* \in K$, the following holds. For each $i = 1, \ldots, m$, pick arbitrarily $T_i > 0$ and $\beta_i^* \in [\beta_i^-, \beta_i^+]$, and let $\beta_i \in L^\infty(\Re_{\geq 0}; [\beta_i^-, \beta_i^+])$ be $T_i$-periodic. Denote by $x$ the solution of Eq (1) associated with the initial condition $x(0) = x_0$ and $\beta_i$. Similarly, denote by $x^*$ the solution of Eq (1) associated with $x^*(0) = x_0^*$ and $\beta_i^*$. Then, for all $t \geq 0$, the following estimate holds:

$$\sup_{0 \leq s \leq t} \|x(s) - x^*(s)\| \leq \alpha_1(t)\mathcal{IC} + \alpha_2(t)\mathcal{P} + \alpha_3(t)\mathcal{A} \tag{11}$$

**Fast open-loop Periodic Switching Policy (FPSP)**

**FPSP (with slow outer-loop supervisory control)**

**Fig 6. Illustrative scheme of the proposed mitigation strategy.** (Green) an example of an FPSP with period $T = 7$ days (one week) is shown on the left in which two consecutive periods with $[X, Y] = [1, 6]$ corresponding to a duty-cycle DC = $1/7 \simeq 14.3\%$ (green box on the left) and a subsequent period with a different FPSP policy $[X, Y] = [2, 5]$ corresponding to a duty-cycle DC = $2/7 \simeq 28.5\%$ are shown (dashed orange box). The application of this FPSP influences the actual dynamics of the epidemics as shown on the right (long left-right arrow on the top). (Orange) delayed and possibly uncertain empirical measurements are collected (green box on the right) and used by the adaptive outer supervisory controller to select the specific FPSP policy (bottom right to left orange arrow) to be used in the subsequent time-period (bottom to top vertical orange arrow pointing to the new FPSP policy in the dashed orange box).

in which the distance $\mathcal{IC}$ between the initial conditions, the maximal period $\mathcal{P}$, and the maximal distance $\mathcal{A}$ between the average value of $\beta_i$ and $\beta_i^*$ are defined as

$$\mathcal{IC} = \|x_0 - x_0^*\|, \quad \mathcal{P} = \max_{1 \le i \le m} T_i, \quad \mathcal{A} = \max_{1 \le i \le m} \left| \frac{1}{T_i} \int_0^{T_i} \beta_i(s) \, \mathrm{d}s - \beta_i^* \right|.$$

◁

**Comment A:** For ease of interpretation, we make the following comments regarding the above results.

A1: If, for each $i = 1, \ldots, m$, $\beta_i$ is $T_i$-periodic and has mean value equal to $\beta_i^*$, then in Eq (11), $\mathcal{A} = 0$. If, moreover, the considered solution obtained with $\beta_i$ and that obtained with $\beta_i^*$ start from the same initial condition (i.e. $x_0 = x_0^*$), then, in Eq (11), $\mathcal{IC} = 0$. In this case, Theorem 1 claims that the distance between the two solutions is proportional only to the maximum period of the functions $\beta_i$ and thus, it can be decreased by increasing the frequency $1/T_i$ of each function $\beta_i$.

A2: In fact, Theorem 1 implies that, as the switching frequency grows to infinity (i.e., as the switching period $T_i$ approaches zero), the time evolution of the epidemic—the dynamics of which is described by Eq (1)—asymptotically approaches the dynamic behaviour of the

*average system*

$$\dot{x}^*(t) = f_0[x^*(t)] + \sum_{i=1}^{m} \beta_i^* f_i[x^*(t)] . \tag{12}$$

It is worth mentioning that this observation can also be obtained from general qualitative results for systems affine in controls. See in particular Theorem 1 in [38]. However, the specific nature of the control policy considered in our framework allows us to derive a quantitative result taking the form of the explicit bound (Eq 11).

A3: Moreover, it is also worth noting that Eq (11) provides a *quantitative estimate* of the discrepancy between dynamics driven by the FPSP and the average one given by Eq (12) for each given switching period $T_i$. In fact, the quantity $\mathcal{A}$ decreases as $T_i$ decreases thus showing the benefits of choosing suitably high frequencies when implementing our FPSP; namely, that fast switching allows us to control the dynamics of the epidemics in a precise manner.

Next, we provide a qualitative analysis of the influence of the initial proportion of infected over the total population $N$ on the peak of the epidemic outbreak. For simplicity, we focus on the fundamental SIR dynamics, given by Eq (1) with the choice (2) for some $\alpha, \sigma > 0$. We recall that in this case $x = (S, I, R)$, in which $S(t), I(t) \in [0, N]$ denote the relative number of susceptible and infected individuals. Moreover, for constant values of $\beta_1(t)$, say $\beta_1(t) = \beta$ for some $\beta \in [0, 1]$, the basic reproduction number reads as $\mathcal{R}_0 = \beta\sigma/\gamma$ (see [3]). If $\mathcal{R}_0 > 1$, the possible spread of the infection in the population depends on the initial number of susceptible $S(0)$. Specifically, a number of susceptible $S(0) \leq N/\mathcal{R}_0$ implies the decrease of the number of infected $I(t)$ to zero. However, a number of susceptible $S(0) > N/\mathcal{R}_0$ implies an initial growth of the epidemic. In such a situation, without switching there exists a (unique) time $t_p$, referred to as the peak time of the epidemic, such that the number of infected $I(t)$ grows on the time interval $[0, t_p]$, and then decreases for $t \geq t_p$ converging to zero. In particular, it can be shown that the maximum number of infected $I_p = I(t_p)$ is given by

$$I_p = N + \frac{N}{\mathcal{R}_0}\left(\log\left(\frac{N}{\mathcal{R}_0 S(0)}\right) - 1\right) .$$

Moreover, a lower bound $t_{p,lb}$ for the peak time $t_p$ is given by

$$t_{p,lb} = \frac{1}{\beta\sigma(1 - \mathcal{R}_0^{-1})} \log\left(\frac{\mathcal{R}_0 I_p}{N} \cdot \frac{S(0)}{I(0)}\right) .$$

**Comment B:** Now, we make two comments with a view to parsing the above result.

B1: It can be seen that the higher the initial number of infected, the higher the peak $I_p$.

B2: From the lower bound $t_{p,lb}$, it can be seen that the smaller $I(0)$, the larger the peak time $t_p$.

These qualitative results confirm that the timing of the initial lockdown is crucial. A prompt lockdown makes it possible to limit the initial number of infected. As a consequence, we can limit and shift in time the peak of infected. This gives more time to develop treatments and vaccines while limiting as much as possible the pressure of the health care system, and allows more time for measurements to be gathered. This latter point may be important for the design of feedback-based mitigation strategies. Moreover, they also may explain the apparent different dynamics in different countries; namely that different countries appear to see a peak in

infectives at very different times, even when the epidemic commenced at roughly the same time.

**Proof of Theorem 1:** For all $t \geq 0$, define

$$B(t) = \max_{1 \leq i \leq m} B_i(t) = \max_{1 \leq i \leq m} \sup_{s \in [0,t]} \left| \int_0^s \{\beta_i(\xi) - \beta_i^*\} \, d\xi \right|. \tag{13}$$

We first show that there exist strictly increasing functions $\alpha_1, \kappa \colon \Re_{\geq 0} \to \Re_{\geq 0}$ such that

$$\sup_{0 \leq s \leq t} \|x(s) - x^*(s)\| \leq \alpha_1(t)\|x_0 - x_0^*\| + \kappa(t)B(t), \quad \forall t \geq 0. \tag{14}$$

Let $\Delta(t) = x(t) - x^*(t)$. From Eq (1) we obtain

$$\dot{\Delta}(t) = \{f_0[x(t)] - f_0[x^*(t)]\} + \sum_{i=1}^m \{\beta_i(t)f_i[x(t)] - \beta_i^* f_i[x^*(t)]\},$$

and thus, with $\Delta_0 = x_0 - x_0^*$, we get

$$\Delta(t) = \Delta_0 + \int_0^t \{f_0[x(s)] - f_0[x^*(s)]\} \, ds + \sum_{i=1}^m \int_0^t \{\beta_i(s)f_i[x(s)] - \beta_i^* f_i[x^*(s)]\} \, ds.$$

As the functions $f_i$ are continuously differentiable and $K$ is compact, the following quantity is well-defined

$$L_i = \max_{x \in K} \left\| \frac{df_i}{dx}(x) \right\|$$

and equals the Lipschitz constant of $f_i$ when restricted to $K$. Moreover, as $K$ is positively invariant for Eq (1) and $x_0, x_0^* \in K$, we obtain that $x(t), x^*(t) \in K$ for all $t \geq 0$. Then,

$$\|\Delta(t)\| \leq \|\Delta_0\| + \int_0^t \|f_0[x(s)] - f_0[x^*(s)]\| \, ds$$

$$+ \sum_{i=1}^m \left\| \int_0^t \{\beta_i(s)f_i[x(s)] - \beta_i^* f_i[x^*(s)]\} \, ds \right\|$$

$$\leq \|\Delta_0\| + L_0 \int_0^t \|\Delta(s)\| \, ds + \sum_{i=1}^m \left\| \int_0^t [\beta_i(s) - \beta_i^*]f_i[x(s)] \, ds \right\|$$

$$+ \sum_{i=1}^m \left\| \int_0^t \beta_i^* \{f_i[x(s)] - f_i[x^*(s)]\} \, ds \right\|$$

$$\leq \|\Delta_0\| + \left\{ L_0 + \sum_{i=1}^m |\beta_i^*| L_i \right\} \int_0^t \|\Delta(s)\| \, ds + \sum_{i=1}^m \left\| \int_0^t [\beta_i(s) - \beta_i^*]f_i[x(s)] \, ds \right\|.$$

We study the last term of the latter inequality. Let $\phi \colon \Re_{\geq 0} \to \Re^n$ be an absolutely continuous function. Then, integrating by parts yields

$$\int_0^t \{\beta_i(s) - \beta_i^*\}\phi(s) \, ds = \int_0^t \{\beta_i(\xi) - \beta_i^*\} \, d\xi \, \phi(t) - \int_0^t \int_0^s \{\beta_i(\xi) - \beta_i^*\} \, d\xi \, \phi'(s) \, ds.$$

Therefore

$$\left\| \int_0^t \{\beta_i(s) - \beta_i^*\}\phi(s)\,\mathrm{d}s \right\| \le \left| \int_0^t \{\beta_i(\xi) - \beta_i^*\}\,\mathrm{d}\xi \right| \|\phi(t)\|$$

$$+ \int_0^t \left| \int_0^s \{\beta_i(\xi) - \beta_i^*\}\,\mathrm{d}\xi \right| \|\phi'(s)\|\,\mathrm{d}s$$

$$\le \left\{ \|\phi(t)\| + \int_0^t \|\phi'(s)\|\,\mathrm{d}s \right\} B_i(t).$$

This shows that

$$\|\Delta(t)\| \le \|\Delta_0\| + \left\{ L_0 + \sum_{i=1}^m |\beta_i^*| L_i \right\} \int_0^t \|\Delta(s)\|\,\mathrm{d}s$$

$$+ \sum_{i=1}^m \left\{ \|f_i[x(t)]\| + \int_0^t \|(f_i \circ x)'(s)\|\,\mathrm{d}s \right\} B_i(t).$$

Recalling that the functions $f_i$ are continuously differentiable on $K$, and $K$ is compact and positively invariant for Eq (1), the following quantity is well-defined

$$F_i = \max_{x \in K} \|f_i(x)\|.$$

In particular, we have $\|f_i[x(t)]\| \le F_i$ and

$$\|(f_i \circ x)'(s)\| = \left\| \frac{\mathrm{d}f_i}{\mathrm{d}x}[x(s)] \left\{ f_0[x(s)] + \sum_{j=1}^m \beta_j(s) f_j[x(s)] \right\} \right\|$$

$$\le L_i \left\{ F_0 + \sum_{j=1}^m \max(|\beta_j^-|, |\beta_j^+|) F_j \right\}.$$

Then, by letting

$$M_0 = L_0 + \sum_{i=1}^m \max(|\beta_i^-|, |\beta_i^+|) L_i, \quad M_1 = \sum_{i=1}^m F_i,$$

$$M_2 = \sum_{i=1}^m L_i \left\{ F_0 + \sum_{j=1}^m \max(|\beta_j^-|, |\beta_j^+|) F_j \right\},$$

we get

$$\|\Delta(t)\| \le \|\Delta_0\| + (M_1 + M_2 t) B(t) + M_0 \int_0^t \|\Delta(s)\|\,\mathrm{d}s.$$

By the Gronwall's inequality we then obtain

$$\|\Delta(t)\| \le \|\Delta_0\| + (M_1 + M_2 t) B(t) + M_0 \int_0^t e^{M_0(t-s)} \{\|\Delta_0\| + (M_1 + M_2 s) B(s)\}\,\mathrm{d}s$$

$$\le \underbrace{e^{M_0 t}}_{=\alpha_1(t)} \|\Delta_0\| + \underbrace{\left\{ \left( M_1 + \frac{M_2}{M_0} \right) e^{M_0 t} - \frac{M_2}{M_0} \right\}}_{=\kappa(t)} B(t).$$

The claimed estimate given in bound ((14)) then follows by noting that $\alpha_1$, $\kappa$ and $B$ are increasing functions. To conclude, it remains to show the existence of a constant $C > 0$ such that

$$B(t) \leq C \max_{1 \leq i \leq m} T_i + t \max_{1 \leq i \leq m} \left| \frac{1}{T_i} \int_0^{T_i} \beta_i(s) \, \mathrm{d}s - \beta_i^* \right|, \quad \forall t \geq 0. \tag{15}$$

Let $t \geq 0$ be arbitrary. For any $0 \leq s \leq t$ we have

$$\left| \int_0^s \{\beta_i(\xi) - \beta_i^*\} \, \mathrm{d}\xi \right| \leq \sum_{k=0}^{\lfloor s/T_i \rfloor - 1} \left| \int_{kT_i}^{(k+1)T_i} \{\beta_i(\xi) - \beta_i^*\} \, \mathrm{d}\xi \right| + \left| \int_{\lfloor s/T_i \rfloor T_i}^s \{\beta_i(\xi) - \beta_i^*\} \, \mathrm{d}\xi \right|.$$

The second term satisfies

$$\left| \int_{\lfloor s/T_i \rfloor T_i}^s \{\beta_i(\xi) - \beta_i^*\} \, \mathrm{d}\xi \right| \leq T_i |\beta_i^+ - \beta_i^-|,$$

while the first term, using the fact that $\beta_i$ is $T_i$ periodic, satisfies

$$\sum_{k=0}^{\lfloor s/T_i \rfloor - 1} \left| \int_{kT_i}^{(k+1)T_i} \{\beta_i(\xi) - \beta_i^*\} \, \mathrm{d}\xi \right| = \sum_{k=0}^{\lfloor s/T_i \rfloor - 1} \left| \int_0^{T_i} \{\beta_i(\xi) - \beta_i^*\} \, \mathrm{d}\xi \right|$$

$$= \left\lfloor \frac{s}{T_i} \right\rfloor T_i \left| \frac{1}{T_i} \int_0^{T_i} \{\beta_i(\xi) - \beta_i^*\} \, \mathrm{d}\xi \right|$$

$$\leq s \left| \frac{1}{T_i} \int_0^{T_i} \beta_i(\xi) \, \mathrm{d}\xi - \beta_i^* \right|$$

$$\leq t \left| \frac{1}{T_i} \int_0^{T_i} \beta_i(\xi) \, \mathrm{d}\xi - \beta_i^* \right|.$$

Using Eq (13) and the three latter estimates, we obtain Inequality (15) by defining

$$C = \max_{1 \leq i \leq m} |\beta_i^+ - \beta_i^-|.$$

The estimate (Eq 11) then follows from Eqs (14) and (15).

## Additional simulations

Here we present additional simulation results on the validation of our FPSP strategy using the SIDARTHE model. As previously stated, the total population is set to $N = 10^7$ and the initially infected population is set approximately to 0.1% of $N$. The simulation setting is the same considered so far including the parameters of the SIDARTHE model. Different simulations are obtained for different values of the period and the duty-cycle. Fig 7 shows the distribution of the maximum peak-values (in percentage) of infected people obtained for each model by each $[X, Y]$ FPSP policy with $X$ and $Y$ ranging from 0 to 14 (i.e., the value of $(100/N) \cdot \sup_{t \geq 50}[I(t) + D(t) + A(t) + R(t) + T(t)]$ for the SIDARTHE model). Fig 8, instead, shows the time instants at which such peaks are attained.

The following findings can be ascertained.

- There is a *stability region*, painted in light blue and located in the bottom-left part of the images, in which the peak values are similar to the one we would observe with a complete lock down, i.e. with any policy in which the number $X$ of work days is set to zero, and the

Peak Value of the infected population [% of 10 millions]

| Quarantine Days \ Work Days | 0 | 1 | 2 | 3 | 4 | 5 | 6 | 7 | 8 | 9 | 10 | 11 | 12 | 13 | 14 |
|---|---|---|---|---|---|---|---|---|---|---|---|---|---|---|---|
| 0 | NaN | 58.98 | 58.98 | 58.98 | 58.98 | 58.98 | 58.98 | 58.98 | 58.98 | 58.98 | 58.98 | 58.98 | 58.98 | 58.98 | 58.98 |
| 1 | 0.7058 | 21.98 | 39.33 | 45.77 | 49.12 | 51.02 | 52.69 | 53.6 | 53.76 | 55.02 | 54.89 | 55.25 | 56.16 | 55.76 | 55.73 |
| 2 | 0.7058 | 1.64 | 22.11 | 33.39 | 39.67 | 43.45 | 46.03 | 47.79 | 49 | 50.96 | 51.02 | 51.85 | 53.38 | 52.81 | 52.57 |
| 3 | 0.7058 | 0.7058 | 9.065 | 22.25 | 30.61 | 36.1 | 39.63 | 42.09 | 44.8 | 46.7 | 47.57 | 48.28 | 50.53 | 50.23 | 49.58 |
| 4 | 0.7058 | 0.7058 | 1.665 | 12.96 | 22.41 | 29.06 | 33.98 | 37.43 | 40.48 | 42.28 | 44.26 | 44.57 | 47.63 | 47.76 | 46.85 |
| 5 | 0.7058 | 0.7058 | 0.7058 | 5.971 | 15.16 | 22.41 | 27.97 | 32.57 | 35.99 | 37.72 | 40.89 | 40.84 | 44.66 | 45.31 | 44.48 |
| 6 | 0.7058 | 0.7058 | 0.7058 | 1.733 | 9.274 | 16.69 | 22.84 | 27.48 | 31.35 | 33.83 | 37.47 | 38.18 | 41.64 | 42.87 | 42.34 |
| 7 | 0.7058 | 0.7058 | 0.7058 | 0.7144 | 4.793 | 11.67 | 17.82 | 22.97 | 26.78 | 30.61 | 34.01 | 35.52 | 38.57 | 40.44 | 40.25 |
| 8 | 0.7059 | 0.7059 | 0.7059 | 0.7144 | 1.882 | 7.566 | 13.65 | 19.01 | 23.41 | 27.34 | 30.54 | 32.88 | 35.49 | 38.02 | 38.21 |
| 9 | 0.7059 | 0.7059 | 0.7059 | 0.7144 | 0.7321 | 4.353 | 9.897 | 15.02 | 19.96 | 24.04 | 27.1 | 30.26 | 32.44 | 35.63 | 36.24 |
| 10 | 0.7059 | 0.7059 | 0.7059 | 0.7144 | 0.7321 | 2.137 | 6.769 | 11.9 | 16.52 | 20.76 | 23.75 | 27.68 | 29.46 | 33.25 | 34.31 |
| 11 | 0.7059 | 0.7059 | 0.7059 | 0.7144 | 0.7321 | 0.9396 | 4.3 | 8.911 | 13.5 | 17.58 | 21.19 | 25.14 | 26.69 | 30.91 | 32.44 |
| 12 | 0.7059 | 0.7059 | 0.7059 | 0.7144 | 0.7321 | 0.7618 | 2.514 | 6.553 | 11 | 15.04 | 18.93 | 22.67 | 24.77 | 28.63 | 30.62 |
| 13 | 0.7059 | 0.7059 | 0.7059 | 0.7144 | 0.7321 | 0.7618 | 1.364 | 4.538 | 8.646 | 12.81 | 16.71 | 20.3 | 22.91 | 26.43 | 28.85 |
| 14 | 0.7059 | 0.7059 | 0.7059 | 0.7144 | 0.7321 | 0.7618 | 0.8388 | 3.026 | 6.648 | 10.69 | 14.58 | 18.04 | 21.11 | 24.33 | 27.14 |

**Fig 7. Percentage of peak infections parametrised by [X, Y] in a population of $10^7$ individuals in the SIDARTHE model.**

peak times are close to the time ($t = 50$ days) in which the policies are started. More precisely, with reference to Fig 8, we observe that a policy [X, Y] belonging to the stability region attains the peak at time $t \leq 50 + X$. This, in turn, implies that the trajectory of the total infected population obtained under these policies starts to decay after the first X days of the first period. We further underline that, although two policies belonging to the stability region have a similar peak value, they may show quite different behaviours.

- There is an *instability region*, painted in dark blue and located in the top-right part of the images, in which the peak values are similar to the one we would observe without lock down, i.e. with any policy in which the number Y of quarantine days is set to zero. As the peak-time distributions in Fig 8 shows, the policies belonging to this instability region do not necessarily lead to the same time evolution. In fact, policies with more days of quarantine (i.e., larger Y) are associated with larger peak times. This, in turn, implies that more days of quarantine still have the positive effect of delaying the peak.

- There is a *compromise region*, which contains the remaining policies, and which is located in the central band going from the top-left to the bottom-right corner. The policies of this region yield a peak of the number of infected people which is considerably larger than the value attained after the initial lockdown phase. However, they are associated with a larger duty-cycle (i.e. a large fraction of work days X) than the policies belonging to the stability region, thus allowing a larger number of work days.

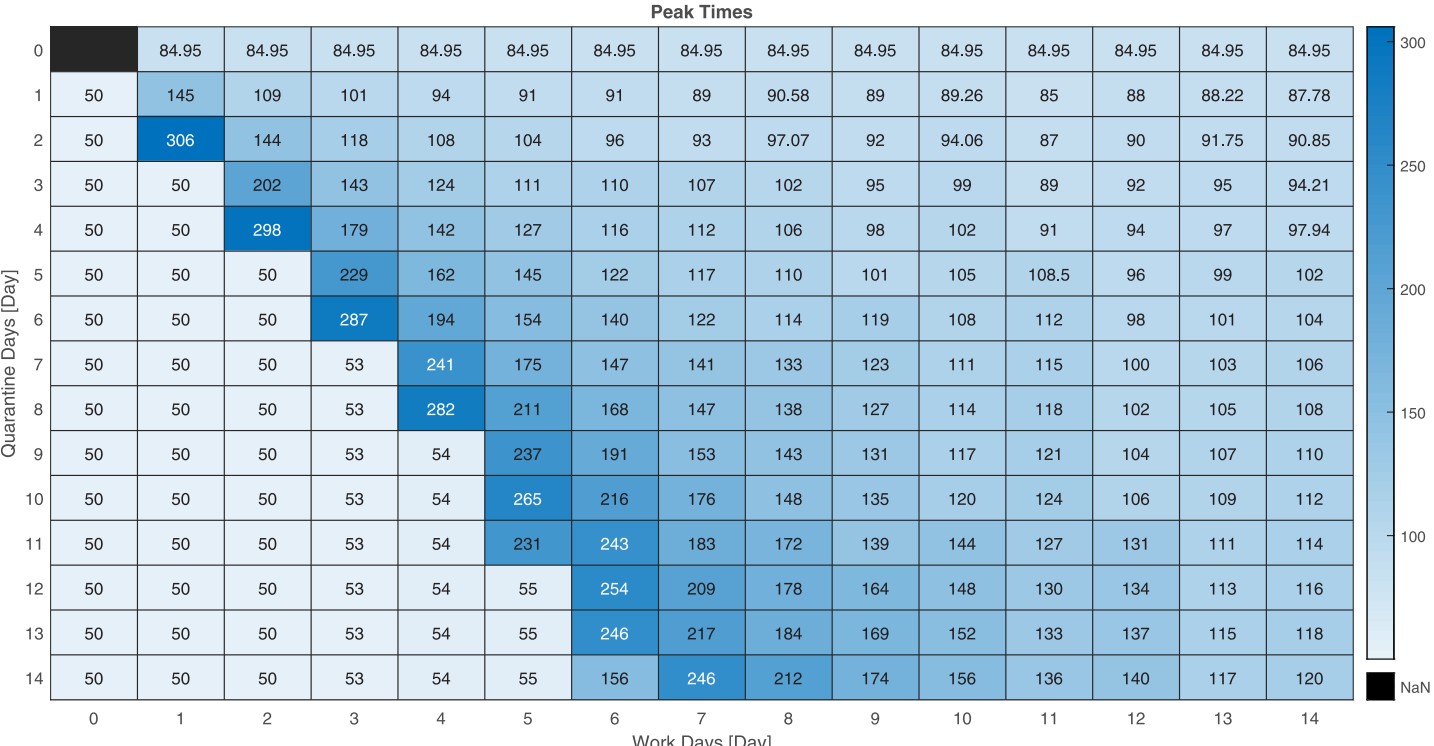

**Fig 8. Distribution of the peak times corresponding to Fig 7.**

## Sensitivity analysis

This section explores the sensitivity of the number of infected individuals in the SIDARTHE model ($I + D + A + R + T$) with respect to quarantine effectiveness, anticipatory and compensatory population behaviour, and uncertainty in the model parameters. Unless stated otherwise, the parameters have the values previously given. An interactive demonstrator that allows the user to change these parameters and observe their effect on an SIQR model controlled by the fast periodic switching policy is available online, linked in the github repository https://github.com/V4p1d/FPSP_Covid19/.

**Quarantine effectiveness.** We present now simulation results obtained by different levels of quarantine effectiveness under the action of a [2, 12] FPSP open-loop policy (Fig 9, top) and of the same policy complemented with the outer loop (Fig 9, bottom). In particular, in what follows we let $\beta^- = q\beta^+$ and we simulate different values of the scaling factor $q$. The open-loop policy remains stabilising for $q \leq 0.335$, corresponding to approximately 66% reduction in the reproduction number during periodic quarantine days. The outer loop guarantees improved stabilising properties also for values of $q$ for which the FPSP open-loop policy is not stabilising (see the case $q = 0.375$). It is worth noting that a further increase in value of $q$ leads to scenarios corresponding to an $\mathcal{R}_0 > 1$ during lockdown periods. In such cases, the outer loop drives the duty-cycle to 0 which corresponds to a full lockdown situation.

**Anticipatory and compensatory population behaviour.** We now explore following situation. We assume that individuals, because they are aware that they will be in lockdown for several days each period, will be more likely to go out and mix during the non-lockdown periods. Thus, in the following situations we augment up $\beta^+$ by a factor $(1 + d)$. Simulation results

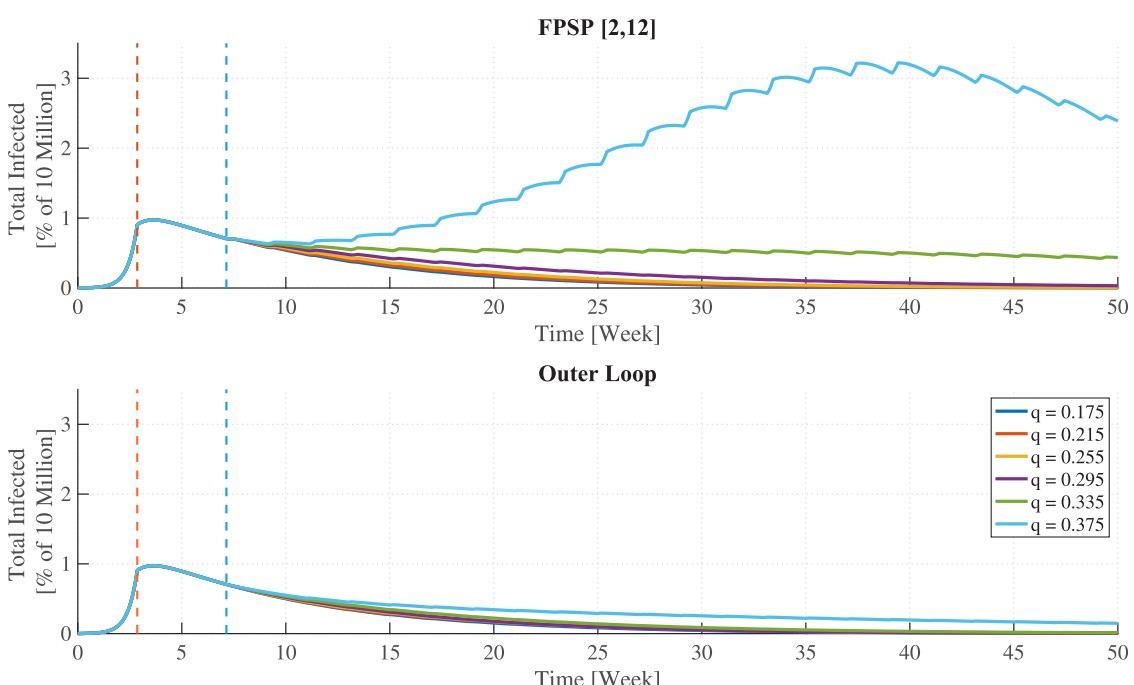

**Fig 9. Sensitivity analysis of the quantity $I + D + A + R + T$ in the SIDARTHE model on quarantine effectiveness $q$. (Top)** The [2, 12] FPSP policy shows a good stabilising behaviour $q \leq 0.335$, corresponding to approximately 66% reduction in infectious contacts during periodic quarantine days whereas a unsatisfactory behaviour is shown for higher values of $q$ (see the trajectory for $q = 0.375$.) **(Bottom)**. The outer loop action is added to the FPSP policy showing a stabilising behaviour also for $q = 0.375$.

are presented modelling increased mixing with a scaling factor $(1 + d)\beta^+ \geq 1$ during working days with the [2, 12] open-loop FPSP policy (Fig 10 top), and complementing this open-loop policy with the outer loop (Fig 10 bottom), respectively. The open-loop policy remains stabilising for $d \leq 0.80$ but fails to stabilise the epidemic for $d = 1$. Instead the outer loop provides improved stabilising properties and also copes well with the case $d = 1$.

**Uncertainty in model parameters.** We consider uncertainty in model parameters represented as zero-truncated Normal distributions with means $\sigma_1, \ldots, \sigma_{16}$ and 10% standard deviation, and estimate $\beta$ from samples of the basic reproduction number $\mathcal{R}_0 \sim \mathcal{N}(\mu = 2.676, \sigma = 0.572)$. By Monte Carlo simulations, 1000 samples were drawn from the joint distribution of the parameters. In order to focus the analysis on the effect of the open-loop FPSP policy, the initial quarantine period for each simulation trial is set to the minimum time for which the infected individuals equals the 2.1% of the total population (i.e., the value estimated for the median number of observed infected individuals after 50 days). The median, 75-percentile and 95-percentile of infected individuals under example policies are shown in Fig 11. Stabilising open-loop FPSP parameterisations avoiding a second peak of infections with 95% probability exist for the SIDARTHE model, both for biweekly ($X \leq 4$) and monthly ($X \leq 8$) switching periods. The outer loop stabilizes all policies including those with period lengths of 16 weeks (Fig 11, right column). The simulations further show that growth trends highlighted above persist across the distribution of model parameters explored here: the peak infected population increases with increasing duty-cycle for fixed period lengths (left to right in each row) and, notably, with increased period length for fixed duty-cycles (top to bottom in each column).

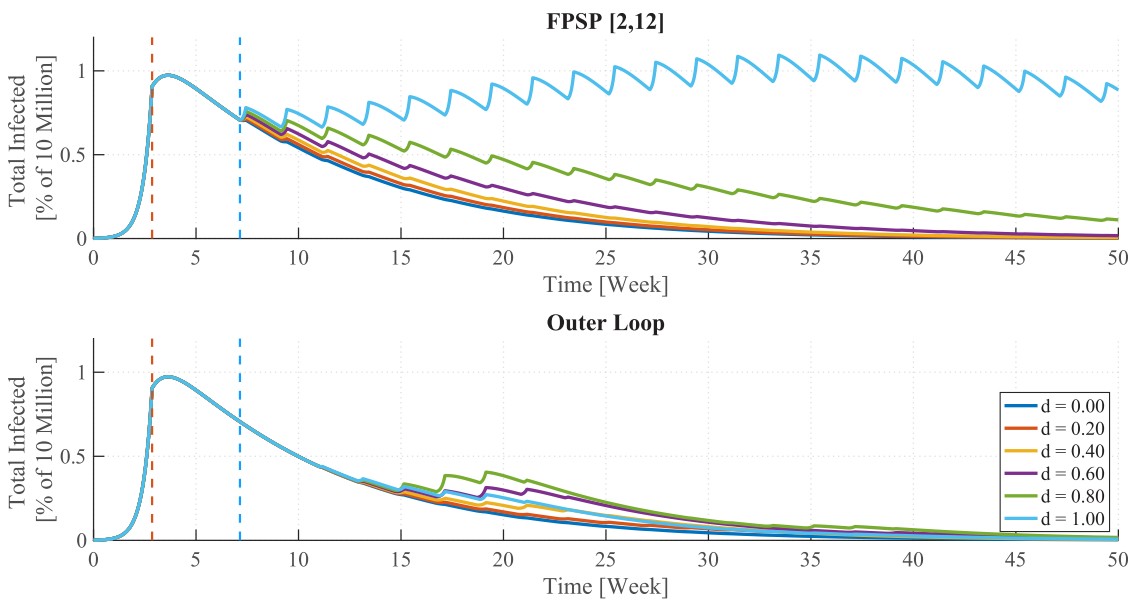

**Fig 10. Sensitivity analysis of the quantity $I + D + A + R + T$ in the SIDARTHE model on increased scaling factor $(1+d)\beta^+$ during periodic working days.** **(Top)** The [2, 12] open-loop FPSP policy remains stabilising for $d \leq 0.80$, corresponding to a 80% increase in infectious contacts during periodic working days due to compensatory and anticipatory population behaviour but fails to stabilise for $d = 1$. **(Bottom)** The outer loop guarantees improved stabilising behaviour also in the case $d = 1$.

## Policy options with FPSP

One can further characterize the trade-off between epidemiological parameters, population behavior, and policy decisions using the notion of level sets. These are parameter configurations that result in an equivalent peak number of simultaneously infected individuals (see Figs 12 and 13). In this manner one may visualise the set of policy interventions that result in similar levels of peak infection outcomes. To this end, we estimate the level sets by fitting a Gaussian Process prior model to the peak infection outcomes of 1000 sample simulations, where the two parameters under investigation were sampled from a Sobol sequence using the software package Ax available from https://ax.dev/, and all others were held fixed at their default values as given above. As we have mentioned, identifying equivalent configurations can help policy makers assess and adjust to the effects of complementary policy decisions and to changes in population behavior. For example, measures such as social distancing, that reduce infectivity, may reduce $\mathcal{R}_0$ sufficiently to safely increase the number of working days per period (Fig 12, left). Conversely, lower compliance with lockdown restrictions, corresponding to increased $q$, may require a decrease in the number of working days to contain the epidemic.

## Unmodelled synchronisation effects

Synchronisation effects may—in principle—affect the design of the virus mitigation strategy described in the paper. In fact, there are several possible pathways for such effects to manifest themselves, all associated with uncertainties. We now briefly comment on these effects and their potential impact on the FPSP. Synchronisation between the disease dynamics and the FPSP policy may emerge due to the following aspects of the disease dynamics.

1. The time between exposure and an individual becoming infective.

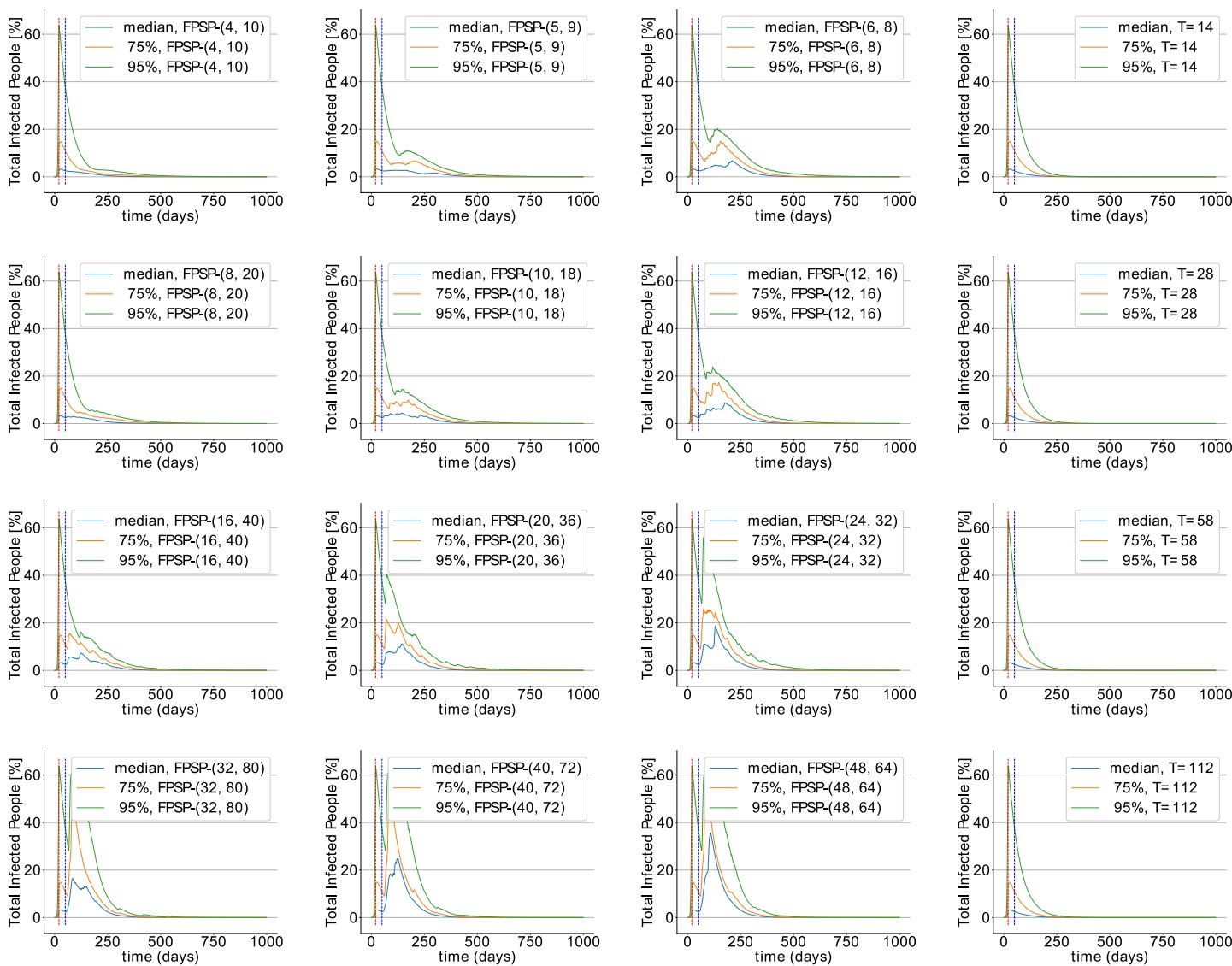

**Fig 11. Sensitivity analysis of the quantity $I + D + A + R + T$ in the SIDARTHE model on uncertainty in epidemiological model parameters.** The y-axis has a linear scale for better visibility of differences across plots. Simulations in each row have fixed period length. Simulations in each column have fixed duty-cycle apart from the right-most column, which shows the effect of the outer loop. Parameters were sampled from zero-truncated Normal distributions with means $\sigma_1, \ldots, \sigma_{16}$ and 10% standard deviation. $\beta$ was estimated from samples of the basic reproduction number $\mathcal{R}_0 \sim \mathcal{N}(\mu = 2.676, \sigma = 0.572)$ representing the consensus distribution in [39].

2. The fact that ODE-based SIR-like models assume a fixed rate of movement from one compartment to another. This is an approximation that corresponds to an exponential distribution. However, recoveries, quarantines, and other quantities, are in reality governed by a more complex distribution.

3. Interactions between the FPSP policy and the behaviour of the population (as individuals may be more likely to go out directly after a lockdown, thereby increasing the level of social mixing).

4. Interactions between the FPSP policy, and possible increase of synchronisation of infections during the open periods.

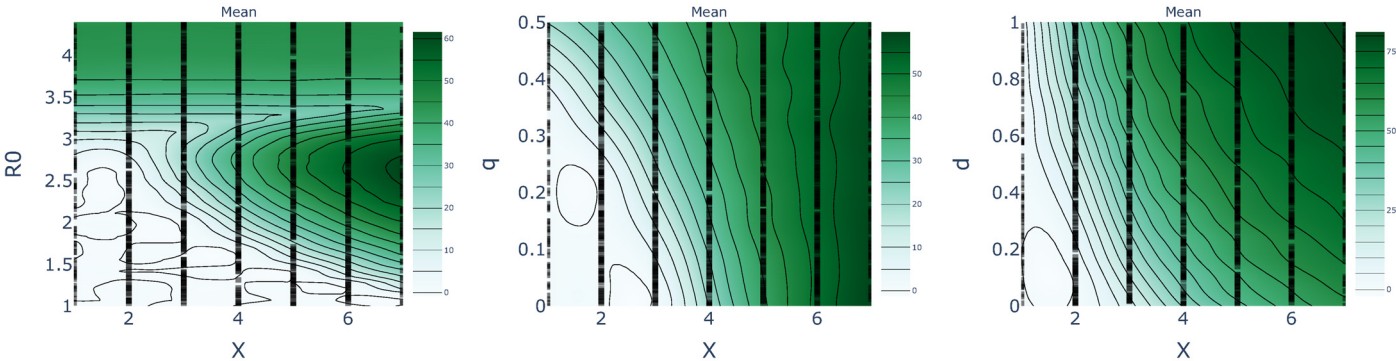

**Fig 12. Level sets as a function of the working days *X* obtained with an FPSP of period 7 days. (Left)** Changing $\mathcal{R}_0$ in [1, 4.4]. **(Centre)** Changing quarantine effectiveness *q* in [0, 1]. **(Right)** Changing compensatory factor *d* in [0, 1].

There are certainly other sources of interaction in addition to these. Our rationale for not directly addressing such effects in our models is related to the following observations. First, their is considerable uncertainty in quantifying them. For example, in reality, there is large uncertainty in quantifying the distribution that governs the transition from an Exposed to Infected class. There is also huge uncertainty in quantifying the distribution governing the movements of individuals from Infected to Quarantine and Susceptible to Infected. Further, the behavioural aspects of how individuals respond to a lockdown is also highly uncertain, and likely to be influenced by many factors, and will also change over time. Second, different simulations, obtained with models having a more realistic distribution of the incubation time and infectivity profile, show that the effects of synchronisation and resonance are negligible. For the above reasons we decided not to model such effects and, rather, to rely on simple but well-accepted models which well-capture the overall qualitative behaviour, and second to incorporate the "outer-loop" strategy to automatically mitigate such effects should they ever appear. In particular, the outer-loop is designed to increase the length of lockdown, relative to open-days, to supress any instability that may arise due to unmodelled effects. Notwithstanding these comments on (the motivation for) the outer loop, we also make the following specific comments with regard to the above points.

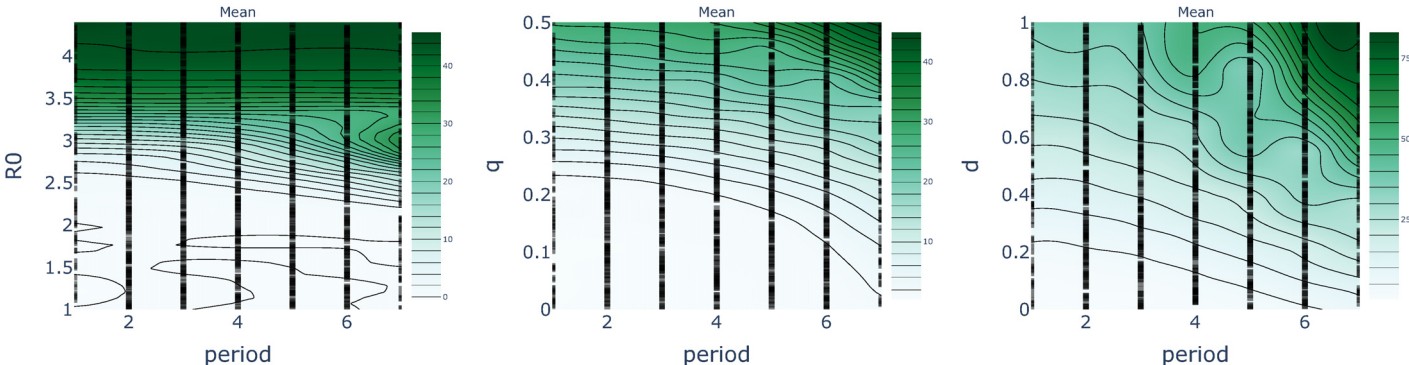

**Fig 13. Level sets as a function of FPSP period *T* and with constant duty cycle DC = 2 / 7. (Left)** Changing $\mathcal{R}_0$ in [1, 4.4]. **(Centre)** Changing quarantine effectiveness *q* in [0, 1]. **(Right)** Changing compensatory factor *d* in [0, 1].

1. Our principal validation tool is the SIDARTHE model which was calibrated using measurements from Lombardy in early Spring, 2020. SIDARTHE corresponds to a strict generalisation of the SIQR model. The disease dynamic is very challenging in the SIQR setting. Here, a susceptible is immediately exposed to infectives, and the disease grows rapidly. This scenario thus provides a robust environment for testing the FPSP policy. However, it should also be noted that the performance of our FPSP policy is not overly sensitive to the specific assumptions of the adopted model, and may be applied in other model settings. For instance we have also tested our policies in SEIQR environments, using stochastic differential equations, as well as agent based simulations, in which the exponential rates are replaced by more realistic distributions, and in which the E class plays a role in the disease dynamics.

2. The assumption of exponential departures and arrivals between classes is, of course, a gross approximation. In addition to incorporating the outer-loop to mitigate this uncertainty, a large part of our existing sensitivity analyses has been concerned with quantifying the impact of more realistic distributions in our models. These are reported in the sensitivity analysis.

3. Finally, in our opinion, the most concerning synchronisation effect may be interactions between the FPSP policy and the behaviour of the population. In this setting, behaviour may drive adaption of the FPSP, and adaption of the FPSP may drive a change in behaviour. There are however, several aspects of the FPSP policy that may serve to dampen the impact of this interaction. First, regular periodic open intervals, with short intervals between these open intervals, may mitigate any interactions due to reduced urgency with which individuals utilise open intervals. Second, during open intervals, we expect both the presence of additional measures, such as the usage of masks and social distancing policies, and a high level of compliance in the population with general health policies to keep the level of mixing to a manageable level. Such levels of compliance, may of course, be enforced by policy and law. Moreover, if people compensate by scheduling all their chores to take place during the non-lockdown days, thus increasing the value of $\beta^+$, then this also implies that they are less active during lockdown days, thus decreasing the value of $\beta^-$. Since it is the average of the two values that counts, this leads to a further damping effect. Finally, interaction of the outer loop with the population is designed in a manner to drive the population towards a full lockdown if negative effects manifest themselves due to increased mixing. Knowledge of this fact may induce or encourage "good behaviour" in the population; if it does not, the unstable interaction between FPSP and the population will drive the system to an equilibrium that supresses the virus anyway (i.e., the full lockdown state). Namely, the outer loop guarantees a safety by enforcing, in the worst case, a full lockdown.

## Discussion and concluding remarks

The main contribution of this paper is to suggest a principled design methodology for designing non-pharmaceutical virus mitigation measures based on regular period switching in and out of lockdowns.

### Main findings

Our main findings may be summarised as follows.

- Our main finding is that a considered design of fast, regular, periodic switching policy (FPSP), in and out of lockdowns over short time-scales, can suppress the COVID-19 outbreak, by regulating the average reproductive number to be below one. This can be achieved

using an open-loop strategy that is not overly dependent on measurements. This strategy can be used until a widespread vaccine becomes available.

- A slow, outer feedback loop, based on averaged measurements, can be used to tune the FPSP, to account for unmodelled effects, and possible synchronisation effects.

- The FPSP policy allows reduced economic and social activity, while at the same time abating the virus. Furthermore, the policy is predictable, and thus more amenable to planned economic activity. This is in contrast to the ad-hoc, data driven, and unpredictable, intermittent lockdowns currently being used to fight the pandemic.

- FPSP complements other virus mitigation strategies such as mask coverings and social distancing.

- The FPSP strategy is based on a rigorous theoretical foundation. Mathematical results are presented that characterise the policy and that can be used to design policies with specific properties. In addition, these policies account for measurement delays that can lead to dynamic instabilities.

## Elaborated discussion

As we write this paper, countries worldwide are experiencing a major resurgence of COVID-19, and many either have or are actively considering reintroducing lockdowns to limit the spread of the virus. It is also understood that the re-introduction of lockdowns will not be easy. Lockdowns place difficult burdens on economies and societies, and the issue of compliance with lockdown policy is likely to be a major societal issue as the disease re-emerges. Thus, in this context, both from a societal and economic perspective, developing strategies of regular activity, followed by short lockdowns, makes sense. As we have shown, such policies can abate the virus to low levels of infectivity, while also allowing regular social and economic activity, and may provide tolerable policies for society for the reintroduction of lockdowns (partial) as countries consider their options in responding to emerging second waves. We also note that COVID-19 is characterised by several uncertainties. Using knowledge from control theory it is possible to account for these delays and uncertainties. In our case, we suggest the use of regular periodic intervention policies that are not overly dependent on real-time measured data. These fast-intermittent exit strategies are robust with respect to uncertainty as lockdown periods are not triggered by measurements, but rather are driven by predictable periodic triggers in- and out- of lockdown. In addition, as we have mentioned over longer periods, uncertain data can be averaged, revealing long-term trends, such as whether mean levels of infections are increasing or decreasing. As some policies are better than others, these can be found by carefully using the averaged data to adjust the specific number of workdays and lockdown days, at a very slow rate, to respond to both uncertainties in the measurements, and changes in the virus dynamics over time.

We note also that classical instabilities are a particular problem in dealing with dynamic systems with delays, and require special treatments of the kind that we advocate to alleviate their effects. This is well known in control theory, but is worth highlighting in the epidemiological context given the significant delays and uncertainties associated with COVID-19. To illustrate this point consider a very simple lockdown algorithm (for example, of the type advocated in [10]) that acts with real time information concerning the epidemic, rather than delayed information. Suppose a lockdown is initiated whenever the number of infected individuals in the population, $I(t)$, exceeds a certain threshold level, say hypothetically $I(t) = 0.002 \cdot$

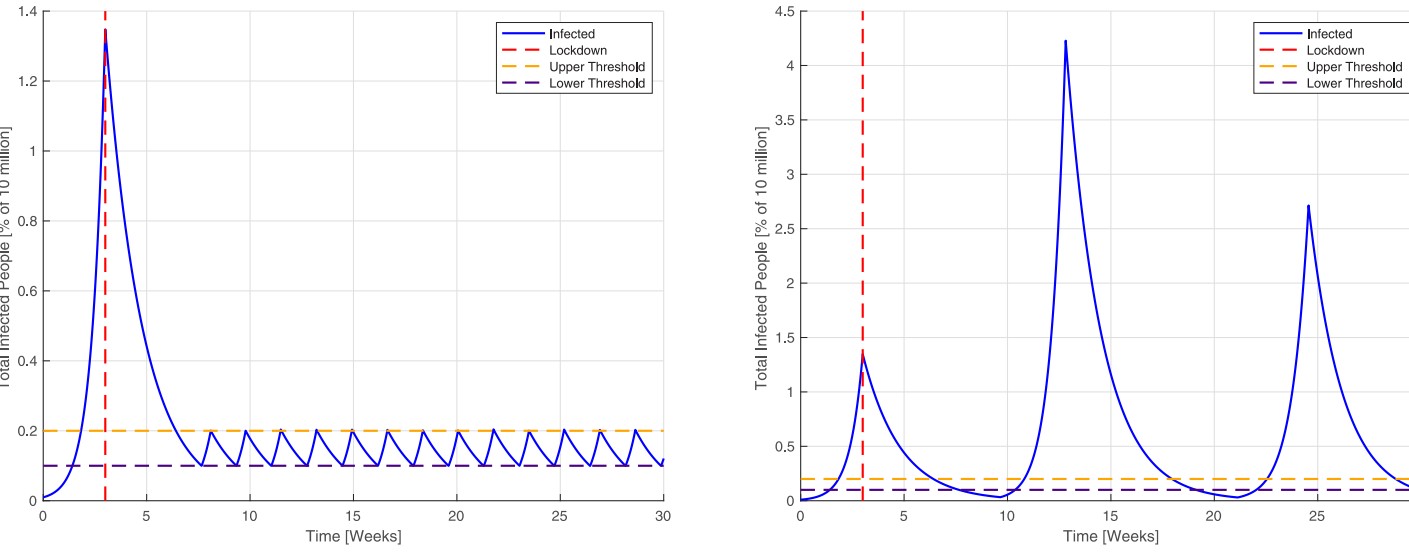

**Fig 14. Threshold-based lockdown using delayed data. (Left)** Time-behaviour of the epidemic modelled by a SIR model under the action of a threshold-based lockdown executed based on data not affected by time-delays. **(Right)** Time-behaviour of the epidemic modelled by the same SIR model under the action of a threshold-based lockdown executed based on data now affected by a 2-weeks time-delay which cause the occurrence of a chain of out-of-control waves capable of infecting $3 - 5\%$ of the population (of the order of 20 times the threshold).

$N$ (i.e., 0.2% of the population). In addition to this, the lockdown is released whenever $I(t)$ falls below another threshold, say $I(t) = 0.001 \cdot N$. Fig 14 (Left) illustrates the effects of this procedure, for demonstration purposes on the most simple SIR model.

The model simulation begins with an epidemic period in which the number of infected individuals $I(t)$ grows exponentially until $t = 3$ weeks, at which point a lockdown is implemented. Because of the lockdown, $\mathcal{R}_0 < 1$, and the epidemic rapidly declines until it falls below the threshold $I(t) = 0.001 \cdot N$ at $t = 8$ weeks and therefore the lockdown is lifted. At this point the control procedure is switched on and the number of infected individuals are regulated so as to bounce between the threshold limits $I(t) = 0.001 \cdot N$ and $I(t) = 0.002 \cdot N$, and are constrained to remain within them. This is the regulation we would like to achieve, and it is achievable when we use real-time data that has no delays, as the SIR model simulation in Fig 14 (Left) shows. However, there are major delays in all aspects of the virus transmission and the detection of infected individuals, and these delays can introduce severe instabilities. The virus itself has an incubation time of 5-7 days before symptoms appear but it can be even up to 14 days. There can quite easily be a week between the time in which the infection symptoms are noted, a test is performed, results analysed, any person who tests positive reaches a hospital bed. Thus the reported confirmed case numbers or the numbers of hospital beds may give a picture of infection events that occurred 2 weeks in the past, or more. Yet the key index for policy makers in many countries is the availability of hospital beds, with the goal of ensuring their availability at all times. The influential paper of Flaxman et al. [1], for example, suggests that the decision to roll out lockdowns should depend on availability of hospital beds. Thus using hospital bed-data as a guide for implementing or releasing a lockdown, implies that decisions are based on the state of the epidemic from several weeks in the past. In Fig 14 (Right), we show the same epidemic outbreak as in Fig 14 (Left) except that the lockdown is switched on or off based on actual data from two weeks in the past. As a result of the time delay, large secondary waves of the epidemic are generated creating havoc in the population with no sign of stability appearing. Thus attempting to control the outbreak with time-delayed

data can easily lead to a large secondary wave, or a chain of out-of-control waves, which is exactly what we were trying to avoid.

Finally, we also mention that it is of course true that social mixing may also be controlled by separating the population into spatial compartments, and that this may be considered an alternative to our proposed strategy. While this is true, there are distinct advantages to the strategy that we are proposing. First, with our strategy, leakage between compartments is easier to manage. Epidemiological dynamics under compartmental population models, with some compartments under lockdown while others are not, are governed by cross-infection rates and therefore depend strongly on the particular choice of population compartmentalisation. Second, and perhaps more importantly, compliance with the strategy is easier to enforce in using a temporal strategy, rather than in a spatial one (as we are now witnessing throughout the world). We believe that a population-wide simultaneous short periodic lockdown, with perhaps essential sectors such as hospitals, transportation sectors, pharmaceutical companies, not locking down at all, is simpler to implement, with respect to public communication, public acceptance, and enforcement, than a permanent lockdown rule based on personal characteristics.

## Concluding remarks

To conclude, the theoretical and empirical simulation results in this paper suggest that policies of switching rapidly in and out-of lockdown, augmented by a slow outer loop, is potentially of great value in abating the effect of COVID-19. While it is beyond the scope of this present paper, initial results have shown the ideas we have developed also perform well in stochastic compartmentalised scenarios, and in agent based models. This latter point is very important, as some compartments in society simply cannot close (hospitals, transport, key industries). Future work will further develop stratified switching strategies, and evaluate the effect of switching in alleviating the need for widespread and targeted COVID-19 testing (sampling) strategies, and will also include cost models to consider these economic effects more explicitly. This latter work is ongoing and will be reported in follow on publications. As a final comment, we note that our proposed strategy is designed to complement, rather than replace other existing viral mitigation strategies, such as social distancing or face coverings.

## Author Contributions

**Investigation:** Michelangelo Bin, Peter Y. K. Cheung, Emanuele Crisostomi, Pietro Ferraro, Hugo Lhachemi, Roderick Murray-Smith, Connor Myant, Thomas Parisini, Robert Shorten, Sebastian Stein, Lewi Stone.

**Methodology:** Michelangelo Bin, Peter Y. K. Cheung, Emanuele Crisostomi, Pietro Ferraro, Hugo Lhachemi, Roderick Murray-Smith, Connor Myant, Thomas Parisini, Robert Shorten, Sebastian Stein, Lewi Stone.

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
