## [Decision Letter · Decision Letter 0]

29 Sep 2020

Dear Professor stone,

Thank you very much for submitting your manuscript "Post-Lockdown Abatement of COVID-19 by Fast Periodic Switching" for consideration at PLOS Computational Biology.

As with all papers reviewed by the journal, your manuscript was reviewed by members of the editorial board and by several independent reviewers. In light of the reviews (below this email), we would like to invite the resubmission of a revised version that takes into account the reviewers' comments. In particular, we would like to see attention to the questions of parameter sensitivity, relative levels of activity, possibility of resonance, and clarification of the results and discussion.  On the conceptual front, referee 2 points out that the main idea and associated results are somewhat "obvious". Can the authors better articulate the ways in which the mathematical/computational analysis contributes in a non-obvious way? 

Please also note that in order to participate in the reproducibility pilot, source code should be provided.

We cannot make any decision about publication until we have seen the revised manuscript and your response to the reviewers' comments. Your revised manuscript is also likely to be sent back to review for further evaluation.

Sincerely,

Mercedes Pascual

Associate Editor

PLOS Computational Biology

Virginia Pitzer

Deputy Editor

PLOS Computational Biology

Reviewer's Responses to Questions

**Comments to the Authors:**

Reviewer #1: Reproducibility report has been uploaded as an attachment.

Reviewer #2: The principle contribution of this work is a mathematical analysis of an extended SIR model (SIDARTHE) with application to a switching parameter space and the (Rather obvious) consequence that the switching between two sets of parameter values effectively acts as a duty cycle. The mathematical analysis is fine, my comments and suggestions related primarily to the simulation and interpretation.

My first and most serious concern is that the paper proposes a switching strategy to exit from a coronavirus induced lockdown. The conclusion - that the net effect is the weighted average of the dynamics during and out of lockdown is reasonable. Although, I would've expected some level of resonance with the incubation period of the disease. However, my issue is that this could easily be misinterpreted by a willing public. That is, the model assume lockdown of n days followed by free movement for m days to allow for partial economic reactivation - although no economic modelling is done. My concern is that the authors assume that the populace will be only as active in the m days of free movement than they would be on a random day without lockdown (and therefore disease spread and economic recovery can both be fractional). A more natural (to me) assumption is that the population will overcompensate during limited periods of "freedom" and exacerbate infection further than predicted here.

My second principle concern is that there is no sensitivity analysis of the presumed parameters of the model.

Minor concerns:

- comparison to season effects is spurious as the frequency is much lower (line 42)

- be -> by( line 66) - there other typos throughout, a careful proofreading would help.

- since the author point out other similar studies [22-24] to lend authority to their model, it would be useful to more thoroughly consider what the particular unique contribution of this work is. (95-97)

- reference to "official statistics" (line 123-4) is confusing. Please say exactly what it is within these statistics that is interesting.

- rather than n-days on and m-days out of lockdown, why not lockdown a fraction of the population? Wouldn't this achieve the same result?

- line 281-2 Eqn (6) : there is a t missing. And, is c distinct from T? Following line, you're assuming that a<c

- line 309-310: why do you need H and E as seperate categories? It serves no purpose here and just complicates the model.

- line 327-333: what effect does these parameter choices have on your results? Where do they come from and do they matter?</c

Reviewer #3: The article is quite interesting and well written. The methodology proposed is clearly presented. The article can still benefit from an extended discussion of the points I list below:

1. Given that the optimal pulsating period for the lock-down is around a week, and the incubation period of the virus is between 1 and 2 weeks, Have you considered the possibility of a resonance between the two oscillations (lockdown and incidence) which could make matters worse? After all, this synchronous release of lock down will force all new infections to happen within a narrow time window creating delayed wave of incidence. This does not show up on your results, but couldn't it be because the SIDARTHE model does not include a E (exposed) compartment? If it did, perhaps the ratio between of the incubation and lockdown periods could become an important parameter to consider.

2. Does working intermittently actual solves the economic problem? I guess the benefits will vary depending on the type of economic activity. For example: for one of the most affected sectors, Tourism, opening for 2 days every week is not sustainable.

Major comments:

Supplementary material is missing, even thought it is referred to in line 668

Figure 10 is missing. without it it is not possible to check the sensitivity of the model to uncertainty in the parameters.

Minor Comments:

line 43: Periodicity is also related to patterns of approaching endemicity, not just disease die-out.

Reviewer #4: The main idea of this article is interesting and could be relevant for policy design. Of course, as the authors point out, this work shows theoretical results and several additional aspects would need to be addressed before application to health strategies.

Basically the proposed idea is to combine regular periods of normal/free and quarantine days. As expected, successful outcomes are those in which the number of normal days X and quarantine days Y give a combined R0 less than 1 (weighted average).

Interestingly, given for example a strategy of X normal days during a month, the incidence would be sensitive to how the X days are arranged in that period of time. That is, the resulting number of infections (as well as peak sizes, etc) would be different if the X days are together than if they are located in a periodic fashion during the month. In particular this work shows that fast periodic switching is better than allocating all the free days together.

It would be valuable for the authors to provide further intuitive explanation for these results, as well as their explicit connection with other systems. For example, when I read the article the first thing that came to mind was the results of periodic forced systems in physics (signals, circuits, etc). It is known that the amplitude of an oscillation will decrease when the forced frequencies are faster than its natural frequency, and that it increases as the forcing frequency gets closer to the natural one. (Does the result on the best X and Y combination depend on N?)

Although the basic idea is nice and “clean”, it took me a while to understand the paper. The reading of the text could be cleared in some places. I provide specific examples for improving the clarity below.

Introduction

- This section is clear but is a bit long and repetitive.

- In some parts, the authors make somewhat strong statements which should be treated with care. See here examples:

- Line 58: high fidelity model? what do you mean? It might be better to avoid terms that can have different interpretations.

- Although the lack of uncertainties in the proposed method is stated in several parts of the text, this is not clear for me. Your results, for example the optimal set of T, X and Y, come from a model whose parametrization is based on clinical datasets (that as you mention have many sources of errors) (moreover, model itself and model parametrization remains a challenge in this area).

Results

- This section is a bit messy and in some places it is difficult to pull out the main message. A better organized version would improve understanding of this section. It may be helpful to add subsections with clear subtitles.

- In addition, it would also be beneficial to explicitly guide the reader through the figures. Figure 5 is a good example of this (only a phrase in the text).

- Figures’ font and legend should be bigger

- Parameters need units (eq 7). For example, beta is a rate (this is specified in line 224 for example), so what does beta=1 mean? if the units are days-1, that would mean that an individual only interacts with one person per day, which is hard to believe. Are g1, g2, g3 and g4 rates (line 311) or probabilities? (g:=sigma). If beta is a rate, then gi, i=1,2,3,4, cannot have units and therefore cannot be a rate. Maybe I am missing something here.

- I understood that these parameters are fitted from clinical data, right? Does this data set correspond to periods with intervention? If so, the model and its parameters represent a situation under intervention. Although the main goal of the paper would be independent of this, your results on the optimal T and the values of beta+ and beta- would not be right? Please, discuss or clarify this.

- Figure 3 is difficult to understand, please consider other options for this.

Discussion

- This section needs to be more straightforward

- When you mention the FPSP for policy making, I would like to see a discussion with other options. For example, your simulations are based on a reduction of almost 85% of the contact rate which is not a small number (moreover, the used beta+ in itself could be quite small; I do not know because I do not know the units) and, in the best of cases for 5 days a week. This corresponds to an R0* of about 0.9. For this scenario, I would like to know your thoughts on the FPSP approach vs a constant “partial-reduced” beta, which for this example, would represent a reduction of 60%. So, is it better to implement 5 days with a reduction of 85% or 7 days with a reduction of 60%?

- Here the uncertainties appear again. Please be clear on what you mean by uncertainties.

- Line 700. Empirical results? I did not see them.

- This is a good place to discuss intuitive explanations of your results.

- There are parts where references are missing and sentences are vague. For example, the phrase that goes from line 638 to 641, ref? Which early results? If you mention this, you do need to explain more clearly what is different now.

- Another example: Line 703: “... are enough to approximate ...” what is enough? Did you define a criteria? I am curious also if the one week is robust under different population sizes and betas.

**Have all data underlying the figures and results presented in the manuscript been provided?**

Reviewer #1: None

Reviewer #2: Yes

Reviewer #3: **No: **

Reviewer #4: Yes

PLOS authors have the option to publish the peer review history of their article (what does this mean?). If published, this will include your full peer review and any attached files.

Reviewer #1: No

Reviewer #2: No

Reviewer #3: **Yes: **Flávio Codeço Coelho

Reviewer #4: No
---

## [Decision Letter · Decision Letter 1]

3 Dec 2020

Dear Professor Stone,

We are pleased to inform you that your manuscript 'Post-Lockdown Abatement of COVID-19 by Fast Periodic Switching' has been provisionally accepted for publication in PLOS Computational Biology.

Best regards,

Mercedes Pascual

Associate Editor

PLOS Computational Biology

Virginia Pitzer

Deputy Editor

PLOS Computational Biology

Reviewer's Responses to Questions

**Comments to the Authors:**

Reviewer #1: Reproducibility report have been uploaded as an attachment.

Reviewer #2: I am happy with the authors responses and revisions and recommend publication.

Reviewer #3: The authors have fully addressed my comments.

Reviewer #4: The authors have addressed all of my comments their revision. I have no further comments.

**Have all data underlying the figures and results presented in the manuscript been provided?**

Reviewer #1: None

Reviewer #2: None

Reviewer #3: Yes

Reviewer #4: Yes

PLOS authors have the option to publish the peer review history of their article (what does this mean?). If published, this will include your full peer review and any attached files.

Reviewer #1: **Yes: **Anand K. Rampadarath

Reviewer #2: No

Reviewer #3: **Yes: **Flavio Codeco Coelho

Reviewer #4: No

---

## [Editor Report · Acceptance letter]

13 Jan 2021

PCOMPBIOL-D-20-01341R1 

Post-Lockdown Abatement of COVID-19 by Fast Periodic Switching

Dear Dr Stone,

I am pleased to inform you that your manuscript has been formally accepted for publication in PLOS Computational Biology. Your manuscript is now with our production department and you will be notified of the publication date in due course.

With kind regards,

Jutka Oroszlan
